# An ACE2 decoy can be administered by inhalation and potently targets omicron variants of SARS-CoV-2

Lianghui Zhang[1,*,‡] (ID), Krishna K Narayanan[2,†] (ID), Laura Cooper[3,†] (ID), Kui K Chan[4], Savanna S Skeeters[4], Christine A Devlin[2] (ID), Aaron Aguhob[4], Kristie Shirley[4], Lijun Rong[3,**] (ID), Jalees Rehman[1,5,***] (ID), Asrar B Malik[1,****] (ID) & Erik Procko[2,4,*****] (ID)

## Abstract

**Monoclonal antibodies targeting the SARS-CoV-2 spike (S) neutralize infection and are efficacious for the treatment of COVID-19. However, SARS-CoV-2 variants, notably sublineages of B.1.1.529/omicron, have emerged that escape antibodies in clinical use. As an alternative, soluble decoy receptors based on the host entry receptor ACE2 broadly bind and block S from SARS-CoV-2 variants and related betacoronaviruses. The high-affinity and catalytically active decoy $sACE_2$.v2.4-IgG1 was previously shown to be effective against SARS-CoV-2 variants when administered intravenously. Here, inhalation of aerosolized $sACE_2$.v2.4-IgG1 increased survival and ameliorated lung injury in K18-hACE2 mice inoculated with P.1/gamma virus. Loss of catalytic activity reduced the decoy's therapeutic efficacy, which was further confirmed by intravenous administration, supporting dual mechanisms of action: direct blocking of S and turnover of ACE2 substrates associated with lung injury and inflammation. Furthermore, $sACE_2$.v2.4-IgG1 tightly binds and neutralizes BA.1, BA.2, and BA.4/BA.5 omicron and protects K18-hACE2 mice inoculated with a high dose of BA.1 omicron virus. Overall, the therapeutic potential of $sACE_2$.v2.4-IgG1 is demonstrated by the inhalation route and broad neutralization potency persists against highly divergent SARS-CoV-2 variants.**

**Keywords** angiotensin-converting enzyme 2; COVID-19; decoy; omicron; receptor; SARS-CoV-2

**Subject Categories** Microbiology, Virology & Host Pathogen Interaction

**EMBO Mol Med (2022) e16109**

## Introduction

Monoclonal antibodies targeting the Spike (S) of severe acute respiratory syndrome coronavirus 2 (SARS-CoV-2) are clinically effective at reducing or preventing COVID-19 symptoms (Gupta *et al*, 2021; Weinreich *et al*, 2021; O'Brien *et al*, 2022; Gottlieb *et al*, 2021). As of June 2022, six antibodies have received emergency use authorization from the U.S. Food and Drug Administration for treating mild-to-moderate COVID-19 (REGN10933/casirivimab, REGN10987/imdevimab (Hansen *et al*, 2020), LY-CoV555/bamlanivimab (Jones *et al*, 2021), LY-CoV016/etesevimab (Shi *et al*, 2020), VIR-7831/sotrovimab (Pinto *et al*, 2020), and most recently LY-CoV1404/bebtelovimab (preprint: Westendorf *et al*, 2022)). Another two antibodies have authorization for prophylactic administration as a slow-release cocktail in immunocompromised patients (AZD8895/tixagevimab and AZD1061/cilgavimab (Zost *et al*, 2020)). All authorized antibodies target the receptor-binding domain (RBD) of the S protein to neutralize infection. While the RBD has the conserved function of binding the human receptor for SARS-CoV-2 cell entry, the RBD sequence is itself poorly conserved across SARS-related betacoronaviruses (Chan *et al*, 2021). Mutational scans have demonstrated that many mutations are tolerated (Starr *et al*, 2020; Chan *et al*, 2021) and the RBD is a region of substantial diversity among SARS-CoV-2 variants in circulation (Hirabara *et al*, 2022). Mutations within the RBD

1 Department of Pharmacology and Regenerative Medicine and the Center for Lung and Vascular Biology, The University of Illinois College of Medicine, Chicago, IL, USA
2 Department of Biochemistry, University of Illinois, Urbana, IL, USA
3 Department of Microbiology and Immunology, The University of Illinois College of Medicine, Chicago, IL, USA
4 Cyrus Biotechnology, Inc., Seattle, WA, USA
5 Department of Biochemistry and Molecular Genetics, The University of Illinois College of Medicine, Chicago, IL, USA
 *Corresponding author. Tel: +1 412 864 5483; E-mail: lhzhang@pitt.edu
 **Corresponding author. Tel: +1 312 355 0203; E-mail: lijun@uic.edu
 ***Corresponding author. Tel: +1 312 996 5552; E-mail: jalees@uic.edu
 ****Corresponding author. Tel: +1 312 996 7636; E-mail: abmalik@uic.edu
 *****Corresponding author. Tel: +1 217 300 1454; E-mail: procko@illinois.edu
 †These authors contributed equally to this work
 ‡Present address: Division of Pulmonary, Allergy and Critical Care Medicine, Department of Medicine, University of Pittsburgh Medical Center, Pittsburgh, PA, USA

allow immune escape and increase transmissibility via enhanced receptor affinity. Rapid viral evolution has been observed after treatment with monoclonal antibody drugs, including the appearance of escape mutations to LY-CoV555 and VIR-7831 in immunocompromised (Jensen et al, 2021) and immunocompetent patients (Rockett et al, 2022). To minimize the likelihood of full escape, noncompeting monoclonal antibodies are combined as cocktails with some success (Baum et al, 2020). For example, the virulent P.1/gamma variant of concern (VOC) carrying 3 mutations in the RBD compared with original virus isolates is resistant to REGN10933 neutralization but is sensitive to REGN10987; the cocktail of the two antibodies remained effective (Copin et al, 2021).

The emergence and rapid spread of the B.1.1.529/omicron VOC have upended the development of monoclonal antibodies for COVID-19. The BA.1 omicron sublineage was first detected in southern Africa in November 2021 and rapidly spread within weeks to displace B.1.617.2/delta as the most prevalent VOC (Viana et al, 2022; Wang & Cheng, 2022). A second omicron sublineage, BA.2, then became dominant (preprint: Lyngse et al, 2022), followed by BA.4 and BA.5 sublineages (preprint: Tegally et al, 2022). Omicron far exceeds other VOCs in its number of mutations; S proteins of BA.1 and BA.2 omicron have approximately 37 and 31 mutations compared with the original virus, of which only 21 mutations are shared by both sublineages (Majumdar & Sarkar, 2022). The RBDs alone, which are targeted by many antibodies, have 15 and 16 mutations, respectively, of which 12 are shared and many are localized to the receptor-binding interface. S proteins of BA.4/BA.5 omicron are identical to BA.2 but carry an additional deletion near the N-terminus and two amino acid substitutions within the RBD, with one more RBD residue reverted to the original sequence. Consequently, there are extensive changes to antigenic epitopes on the surface of S. Neutralizing antibody titers are diminished in the serum of recovered and vaccinated individuals (Cele et al, 2021; preprint: Ikemura et al, 2022; Planas et al, 2022; Rössler et al, 2022), and neutralizing antibody titers to BA.4/BA.5 omicron are even low in individuals who recovered from infection with the earlier BA.1 variant (Cao et al, 2022b). There is further potential for S of omicron variants to continue mutating and escape antibody neutralization (Cao et al, 2022b). BA.1 omicron is reported to escape the REGN10933 + REGN10987, LY-CoV555 + LY-CoV016, and AZD1061 + AZD8895 cocktails (preprint: Ikemura et al, 2022; Planas et al, 2022; VanBlargan et al, 2022; Cao et al, 2022a) and VIR-7831 has markedly reduced efficacy against pseudovirus expressing S of BA.2 omicron, raising the possibility that only one authorized antibody (LY-CoV1404) may remain clinically effective (preprint: Iketani et al, 2022; preprint: Zhou et al, 2022; Cao et al, 2022b). Indeed, in one study, just 2 antibodies from a panel of 19 in preclinical and clinical development remained potent against BA.2 omicron (preprint: Iketani et al, 2022). These findings challenge whether monoclonal antibodies are suitable over the long term for the treatment of endemic COVID-19 without constant updating.

An alternative approach is to use soluble decoy receptors that bind and block the RBD (Hofmann et al, 2004; Chan et al, 2020; Lei et al, 2020; Monteil et al, 2020; Jing & Procko, 2021). S binds to angiotensin-converting enzyme 2 (ACE2), which is highly expressed on type II alveolar lung epithelium (Hamming et al, 2004; Zhao et al, 2020), triggering conformational changes that facilitate the fusion of the viral envelope and host cell membrane (Huang et al, 2020). ACE2 is a protease of the renin–angiotensin–aldosterone system (RAAS), which catalyzes the cleavage and inactivation of the vasoconstrictor angiotensin II (Ang-II) (Tipnis et al, 2000; Vickers et al, 2002), as well as the cleavage of other proinflammatory peptides in the kinin system that promote vascular leakage (Vickers et al, 2002), and formation of Ang 1–7 to mediate anti-inflammation, anti-fibrosis, and vasodilation through Mas signaling (Santos et al, 2003; Kuba et al, 2021). The extracellular domains of ACE2 can be expressed as a soluble protein (sACE2) that blocks the RBD (Hofmann et al, 2004). Recombinant sACE2 has been evaluated in hospitalized COVID-19 patients, where it was found to decrease time on mechanical ventilation but had no positive impact on survival (ClinicalTrials.gov Identifier NCT04335136). To improve efficacy, next-generation sACE2 decoys have been engineered for exceptionally tight affinity to S ($K_D < 1$ nM), on par with monoclonal antibodies (Chan et al, 2020; preprint: Cohen-Dvashi et al, 2020; Glasgow et al, 2020; Higuchi et al, 2021; Sims et al, 2021).

ACE2-based decoys have two proposed advantages that distinguish them from antibodies. First, infection reduces ACE2 activity in the lungs due to cell death and shedding of cellular ACE2 through the action of proteases (Kuba et al, 2005; Haga et al, 2008; Heurich et al, 2014). This causes massive dysregulation of the RAAS, with large increases in serum Ang-II and serum sACE2 (Liu et al, 2020; Wu et al, 2020; Filbin et al, 2021; Kragstrup et al, 2021; Lundström et al, 2021; Reindl-Schwaighofer et al, 2021; Fagyas et al, 2022), although much of the serum sACE2 may have low catalytic activity as well as low S affinity and avidity due to proteolysis within the ACE2 collectrin-like dimerization domain. These serum markers are highly correlated with disease severity and elevated Ang-II may contribute to vasoconstriction, thrombophilia, microthrombosis, and respiratory failure. Administering catalytically active sACE2 can dampen Ang-II and kinin signaling to reduce lung injury (Imai et al, 2005; Treml et al, 2010; Zou et al, 2014; Bastolla et al, 2022). However, many groups have argued that sACE2 therapeutics must be catalytically inactivated to prevent off-target toxicity (preprint: Cohen-Dvashi et al, 2020; Glasgow et al, 2020; preprint: Iwanaga et al, 2020; Lei et al, 2020; preprint: Chen et al, 2021; Higuchi et al, 2021; Sims et al, 2021; Tanaka et al, 2021).

The second proposed advantage is that receptor decoys will have unparalleled breadth against variants of S (Chan et al, 2020, 2021). If a mutant S protein has diminished affinity for the decoy, it will likely have diminished affinity for the native receptor and the virus will be attenuated. Engineered sACE2 decoys tightly bind diverse SARS-CoV-2 variants and related betacoronaviruses from bats (Chan et al, 2021; Higuchi et al, 2021; preprint: Yao et al, 2021; Zhang et al, 2022). Potent neutralization persists against BA.1 omicron for at least some engineered decoys (preprint: Ikemura et al, 2022).

Here, we evaluate these two proposed advantages of an engineered sACE2 decoy and demonstrate that ACE2 catalytic activity contributes to therapeutic efficacy and tight binding persists against S proteins of BA.1, BA.2, and BA.4/BA.5 omicron. We further show that the decoy can be aerosolized and is therapeutically effective via inhalation.

# Results

## sACE2₂.v2.4-IgG1 inhalation alleviates lung injury and increases survival of gamma infected mice

The engineered decoy sACE2₂.v2.4-IgG1 has three substitutions compared with wild-type ACE2 that enhance affinity for S by over an order of magnitude, measured against S proteins from SARS-CoV-2 variants predating omicron (Chan *et al*, 2020, 2021; Zhang *et al*, 2022). Fusion to the Fc region of IgG1 increases serum stability and virus clearance (preprint: Chen *et al*, 2021). We recently found that intravenous (IV) administration of sACE2₂.v2.4-IgG1 mitigates lung vascular endothelial injury and increases survival in K18-hACE2 transgenic mice infected with SARS-CoV-2 variants (Zhang *et al*, 2022). To further characterize the translational potential of sACE2₂.v2.4-IgG1, the protein was nebulized and administered by inhalation to K18-hACE2 transgenic mice. Mice were

inoculated with a lethal dose of virulent SARS-CoV-2 isolate Japan/TY7-503/2021 (P.1/gamma variant) at $1 \times 10^4$ plaque-forming units (PFU) to induce severe lung injury (Zhang *et al*, 2022). sACE2₂.v2.4-IgG1 (7.5 ml at 8.3 mg/ml in PBS) was aerosolized by a nebulizer and delivered to the mice starting from 12 h postinoculation. 3 doses were given 36 h apart (Fig 1A) and aerosol delivery of PBS was applied as a control. The doses and inhalation interval of sACE2₂.v2.4-IgG1 were based on previously published pharmacokinetic studies using inhalation and direct intratracheal delivery (Zhang *et al*, 2022). We estimate the inhaled dose of sACE2₂.v2.4-IgG1 to the lungs to be ~0.5 mg/kg/dose.

We chose the gamma variant to assess the efficacy of the inhaled sACE2₂.v2.4-IgG1 decoy because this variant has been shown to induce severe forms of COVID-19 resulting in high mortality (Lin *et al*, 2021; Zhang *et al*, 2022). In our experimental model, all mice inoculated with the gamma variant and receiving PBS inhalation as a control died at 6–7 days (Fig 1B) with a 30% weight loss (Fig 1C). In

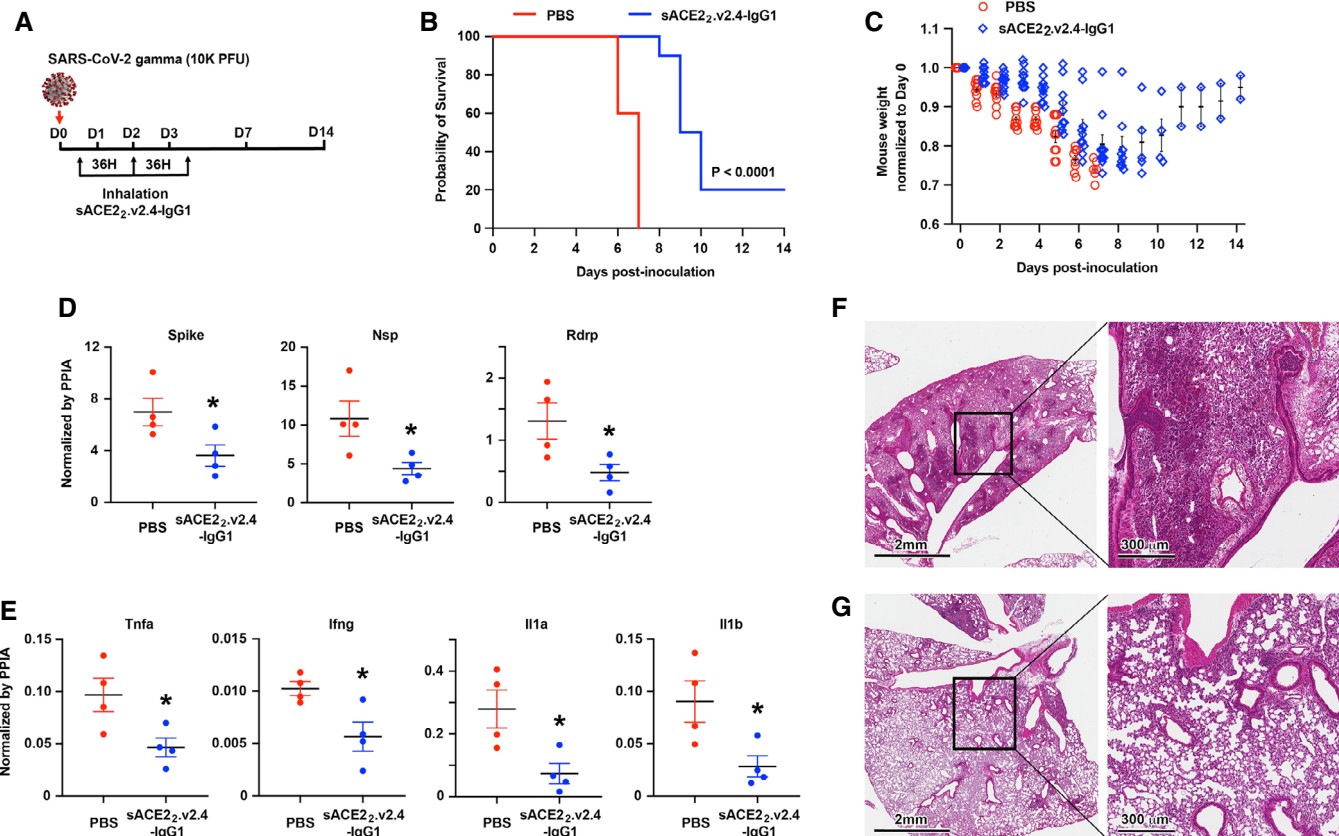

**Figure 1. Aerosol delivery of sACE2₂.v2.4-IgG1 alleviates lung injury and improves survival of SARS-CoV-2 gamma variant infected K18-hACE2 transgenic mice.**

A K18-hACE2 transgenic mice were inoculated with SARS-CoV-2 isolate /Japan/TY7-503/2021 (gamma variant) at $1 \times 10^4$ PFU. sACE2₂.v2.4-IgG1 (7.5 ml at 8.3 mg/ml in PBS) was delivered to the mice by a nebulizer in 25 min at 12 h, 48 h, and 84 h postinoculation. PBS was aerosol delivered as control.

B, C Survival (B) and weight loss (C). *N* = 10 mice for each group. The P-value of the survival curve by the Gehan–Breslow–Wilcoxon test is shown. Error bars for mouse weight are centered on the mean and show SEM.

D Viral load in the lung was measured by RT–qPCR on Day 7. The mRNA expression levels of SARS-CoV-2 Spike, Nsp, and Rdrp are normalized to the housekeeping gene peptidylprolyl isomerase A (Ppia). Data are presented as mean ± SEM, *N* = 4 mice per group. *P < 0.05 by the unpaired Student's *t*-test with two-sided.

E Cytokine expression levels of Tnfa, Ifng, Il1a, and Il1b were measured by RT–qPCR normalized by Ppia. Data are presented as mean ± SEM, *N* = 4 mice per group. *P < 0.05 by the unpaired Student's *t*-test with two-sided.

F, G Representative H&E staining of lung sections on Day 7 postinoculation for control PBS group (F) and inhalation of the sACE2₂.v2.4-IgG1 group (G). Images at left are low magnifications. Boxed regions (black) are shown at higher magnification on the right. Lungs from 4 independent mice were sectioned, stained, and imaged.

the treatment group with inhalation of sACE2.v2.4-IgG1, 20% of mice survived SARS-CoV-2 gamma infection with 5%-15% weight loss and 80% of mice died in 8–10 days, indicating prolonged survival (Fig 1B and C). Replication of SARS-CoV-2 gamma variant in the lung on Day 7 postinoculation was measured by reverse transcription and real-time quantitative PCR (RT–qPCR) for the expression levels of SARS-CoV-2 Spike, nonstructural protein (Nsp), and RNA-dependent RNA polymerase (Rdrp) (Fig 1D). Viral transcript levels are expected to correlate with virus titer. We found that sACE2.v2.4-IgG1 inhalation significantly inhibited viral replication in the lungs. Furthermore, we measured the expression levels of cytokines Tumor necrosis factor alpha (Tnfa), Interferon gamma (Ifng), Interleukin 1 alpha (Il1a), and Interleukin 1 beta (Il1b) in the lungs by RT–qPCR (Fig 1E). Aerosol delivery of sACE2.v2.4-IgG1 reduced cytokine expression in the lungs, although the levels remained higher than in uninfected mice (Table 1). Hematoxylin–eosin (H&E) staining of lung sections demonstrated severe immune cell infiltration induced by gamma variant infection on Day 7 in the control group (Fig 1F). sACE2.v2.4-IgG1 inhalation significantly reduced immune cell infiltration on Day 7 (Fig 1G). Combining all, we conclude that sACE2.v2.4-IgG1 inhalation efficiently alleviated severe lung injury and increased survival following a lethal dose of SARS-CoV-2 gamma variant infection by inhibiting viral replication and reducing cytokine release in the lung.

## Catalytic activity of sACE2.v2.4-IgG1 contributes to therapeutic efficacy

We have previously shown that administration of sACE2.v2.4-IgG1 to mice is associated with large increases in ACE2 catalytic activity detected in plasma (Zhang et al, 2022). To test whether the decoy's proteolytic activity contributes to the mechanism of reducing disease

following SARS-CoV-2 infection, two mutations (H374N and H378N, abbreviated "NN") were introduced to the sACE2.v2.4-IgG1 active site. These mutations disrupt the coordination site of an essential $Zn^{2+}$ ion and were previously shown to have no impact on SARS-CoV-1 infection (Moore et al, 2004). We confirmed that catalytically dead sACE2.v2.4(NN)-IgG1 failed to cleave a substrate peptide (Fig 2A) while its affinity for gamma RBD was unchanged (Fig 2B and C).

We tested the therapeutic efficacy of catalytically dead sACE2.v2.4 (NN)-IgG1 directly against catalytically active sACE2.v2.4-IgG1 using a lethal dose ($1 \times 10^4$ PFU) of SARS-CoV-2 gamma variant to infect K18-hACE2 transgenic mice, in which the proteins were aerosolized and administered via inhalation route as described above. Consistent with its high affinity for blocking S and neutralizing infection, catalytically dead sACE2.v2.4(NN)-IgG1 prolonged survival with 10% survival rate (Fig 2D). However, catalytically active sACE2.v2.4-IgG1 extended survival further by ~1 day longer than catalytically dead sACE2.v2.4(NN)-IgG1 with 20% survival rate (Fig 2D), supporting the hypothesis that ACE2 catalytic activity contributes to therapeutic efficacy. The mice that inhaled catalytically dead sACE2.v2.4(NN)-IgG1 lost more body weight than mice that inhaled catalytically active sACE2.v2.4-IgG1 (Fig 2E). However, no difference was observed in viral load in the lungs on Day 7 between the two groups (Fig 2F), consistent with catalytically dead sACE2.v2.4(NN)-IgG1 neutralizing virus despite having reduced therapeutic efficacy.

To further evaluate the therapeutic contributions of ACE2 catalytic activity, K18-hACE2 transgenic mice were intranasally inoculated with a lethal dose ($1 \times 10^4$ PFU) of SARS-CoV-2 isolate USA-WA1/2020 (an original variant) and protein administration was delayed for 24 h to allow initial virus replication (Fig 3A). This mouse model of COVID-19 induces severe lung injury and edema (Zhang et al, 2022) and was thus considered a challenging test for how catalytic activity may ameliorate disease pathology. Inoculated mice (8/9 per group) were treated with 15 mg/kg/day for 7 days intravenously (Fig 3A). All mice administered an unrelated isotype-matched IgG1 as a control died within 6–7 days, whereas 50% of mice treated with catalytically active sACE2.v2.4-IgG1 survived (Fig 3B and C). These results closely match prior published data in which PBS was administered as the control (Zhang et al, 2022). By comparison, catalytically dead sACE2.v2.4(NN)-IgG1 prolonged survival by just ~1 day with a low survival rate of 11% (Fig 3B and C). This was despite catalytically active sACE2.v2.4-IgG1 and catalytically dead sACE2.v2.4(NN)-IgG1 binding original RBD with equivalent affinities (Fig 3D and E).

Overall, we conclude that sACE2.v2.4-IgG1 has dual mechanisms of action: (i) blockade of receptor-binding sites on SARS-CoV-2 spikes and (ii) turnover of vasoconstrictive and pro-inflammatory peptides that otherwise contribute to lung injury. This conclusion agrees with seminal research demonstrating that ACE2 protects against lung injury (Imai et al, 2005; Treml et al, 2010; Zou et al, 2014) and is also supported by the observation that a bacterial ACE2 homolog protects SARS-CoV-2 infected animals, despite having no affinity for S (Yamaguchi et al, 2021).

## sACE2.v2.4-IgG1 tightly binds and neutralizes BA.1 omicron virus *in vitro* and *in vivo*

Mature ACE2 is composed of a protease domain (amino acids [a.a.] 18-615) that contains the S interaction site, a collectrin-like

**Table 1. Cytokine expression (transcript levels relative to Ppia).**

| | | Day 7 postinoculation | |
|---|---|---|---|
| Gene | Uninfected and untreated | PBS | sACE2.v2.4-IgG1 |
| Il1a | 0.0379 | 0.1985 | 0.1648 |
| | 0.0105 | 0.1562 | 0.0475 |
| | 0.0051 | 0.4054 | 0.067 |
| | 0.0079 | 0.3578 | 0.0165 |
| Il1b | 0.0076 | 0.067 | 0.0248 |
| | 0.0094 | 0.0496 | 0.0183 |
| | 0.0085 | 0.1369 | 0.0126 |
| | 0.0026 | 0.1081 | 0.058 |
| Tnfa | 0.0070 | 0.0854 | 0.0434 |
| | 0.0031 | 0.1345 | 0.0701 |
| | 0.0055 | 0.0595 | 0.0468 |
| | 0.0050 | 0.1083 | 0.026 |
| Ifng | 0.0016 | 0.0095 | 0.0059 |
| | 0.0003 | 0.0108 | 0.0024 |
| | 0.0013 | 0.0089 | 0.0092 |
| | 0.0009 | 0.0118 | 0.0052 |

Lung tissue was harvested from $N = 4$ mice in each group. Expression levels based on RT–qPCR are listed for each animal.

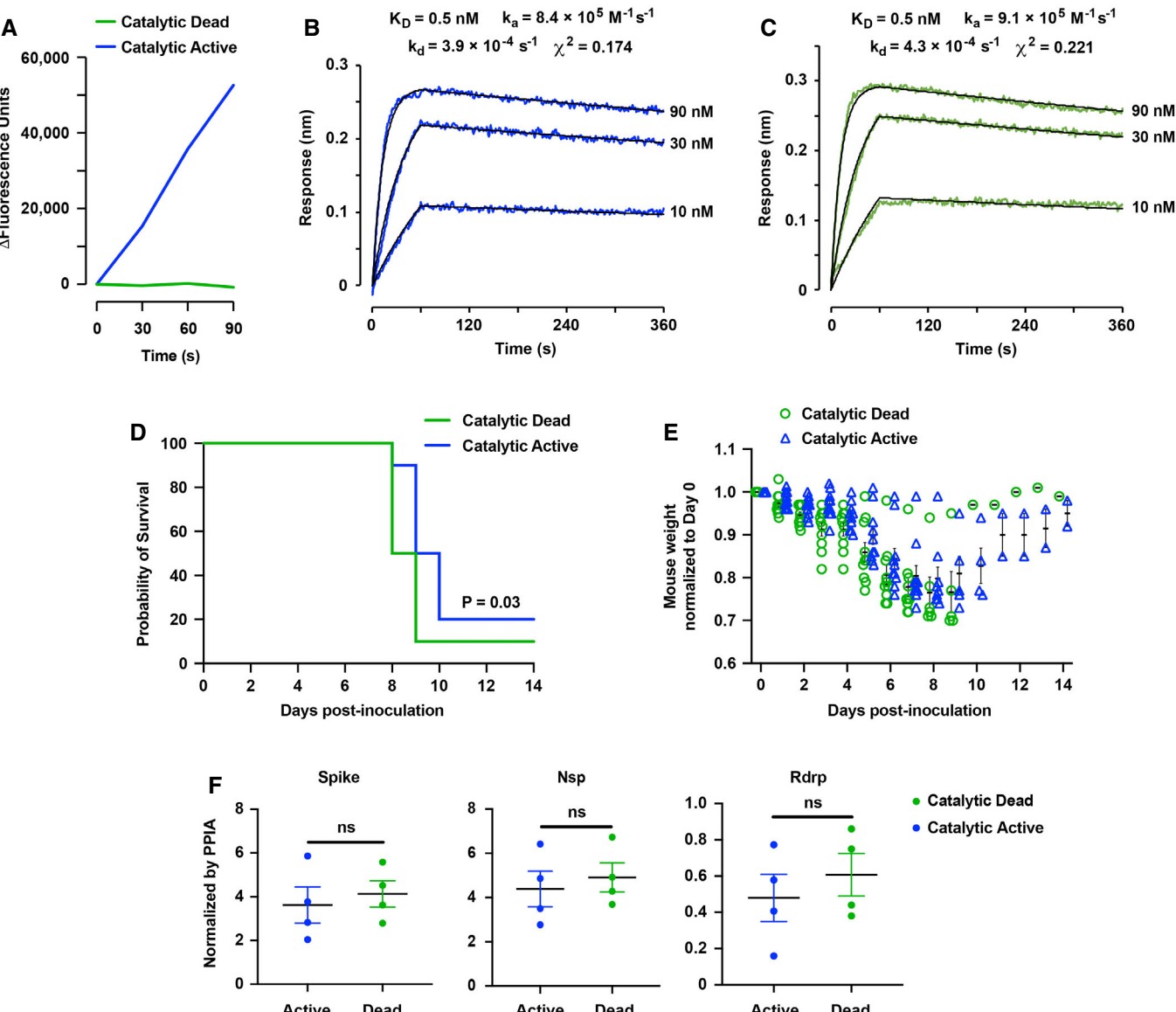

**Figure 2. Catalytic activity of sACE2.v2.4-IgG1 contributes to the therapeutic efficacy to mitigate mouse lung injury and improve survival following SARS-CoV-2 gamma infection.**

A    A 7-Methoxycoumarin-4-acetyl (MCA) conjugated peptide is quenched by a 2,4-dinitrophenyl group. ACE2-catalyzed cleavage of the peptide is measured by increased MCA fluorescence. Mutations H374N and H378N generate a catalytically dead sACE2.v2.4-IgG1 protein.

B, C    Catalytically active sACE2.v2.4-IgG1 (B) and catalytically dead sACE2.v2.4(NN)-IgG1 (C) were immobilized on BLI biosensors that were transferred to solutions of gamma RBD as the soluble analyte (0–60 s) and returned to buffer to measure dissociation (60–360 s). RBD concentrations are indicated on the right of the sensorgrams.

D, E    Catalytically active sACE2.v2.4-IgG1 and catalytically dead sACE2.v2.4(NN)-IgG1 were aerosolized (7.5 ml protein at 8.3 mg/ml in 25 min) and delivered by inhalation to K18-hACE2 transgenic mice at 12 h, 48 h, and 84 h postinoculation with SARS-CoV-2 gamma variant. 10 mice in each group were observed for survival (D) and weight loss (E). The *P*-value of the survival curve by the Gehan–Breslow–Wilcoxon test is shown. Error bars for mouse weight are centered on the mean and show SEM. Catalytically active and inactive proteins were tested in the same experiment versus PBS control shown in Fig 1.

F    Viral load in the lung was measured by RT–qPCR on Day 7. The mRNA expression levels of SARS-CoV-2 Spike, Nsp, and Rdrp are normalized to Ppia. Data are presented as mean ± SEM, *N* = 4 mice per group. ns, no significance by the unpaired Student's *t*-test with two-sided.

dimerization domain (a.a. 616-732), a transmembrane domain (a.a. 741-762), and cytoplasmic tail (a.a. 763-805) (Yan *et al*, 2020). Soluble ACE2 from residues 18-615 is a monomeric protein and its binding to S-expressing cells is dependent on monovalent affinity. Soluble ACE2 from residues 18-732 is a stable dimer (which we

denote as sACE2₂) that binds avidly to S-expressing cells. Avid binding can mask differences in monovalent affinity (Chan *et al*, 2020, 2021; Zhang *et al*, 2022).

We incubated cells expressing BA.1 omicron S with three monomeric sACE2(18-615) proteins: wild-type, v2.4 (ACE2 mutations

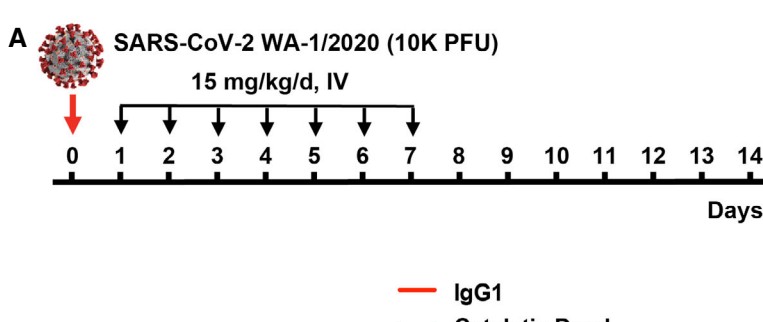

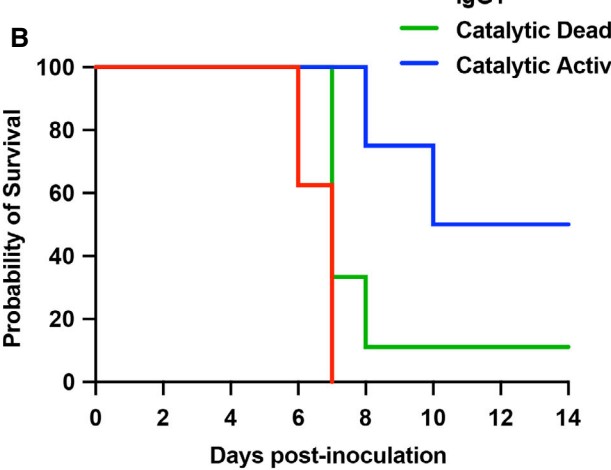

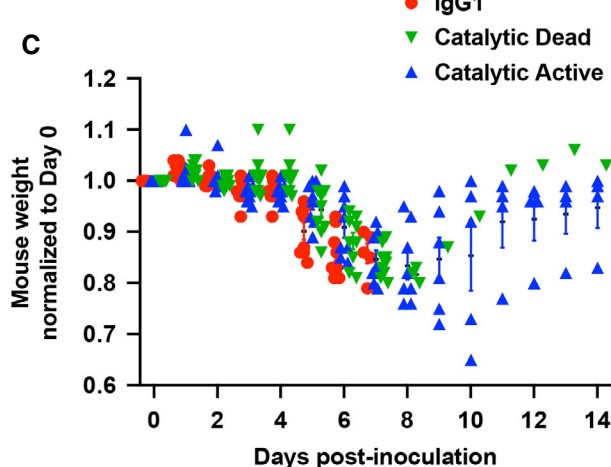

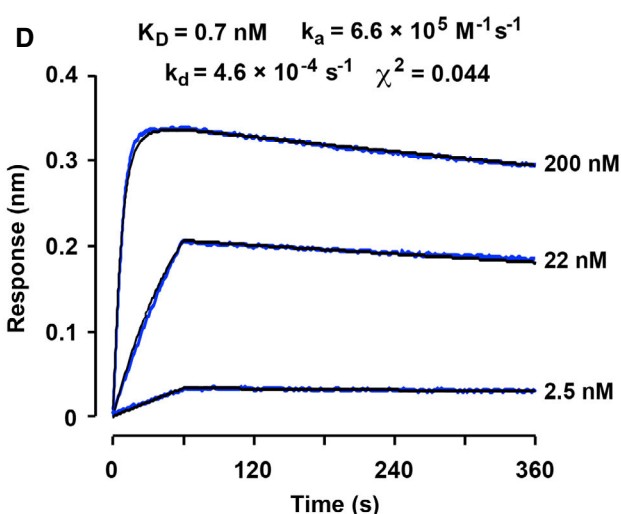

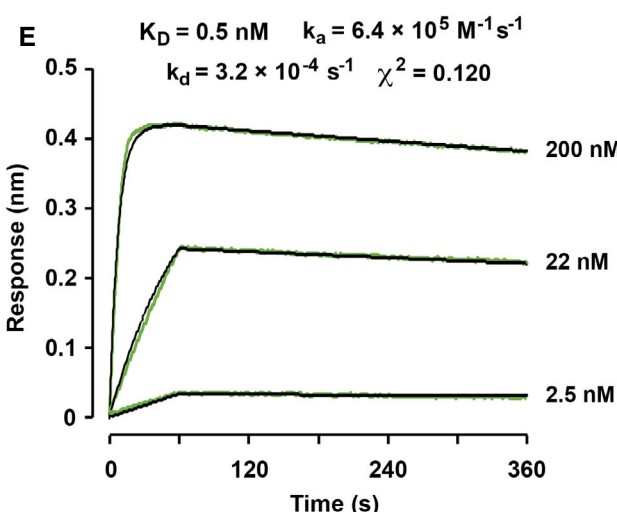

**Figure 3. sACE2.v2.4-IgG1 catalytic activity increases survival of SARS-CoV-2 WA-1/2020 infected K18-hACE2 mice.**

A    K18-hACE2 transgenic mice were inoculated with $1 \times 10^4$ PFU SARS-CoV-2 WA-1/2020 (original variant). Treatment was delayed by 24 h to allow initial virus replication and evaluate the effects of ACE2 catalytic activity at later stages of the disease. sACE2.v2.4-IgG1 was intravenously administered at 15 mg/kg, once per day for 7 days.

B, C    Survival (B) and weight loss (C) following treatment with a IgG1 isotype control (red) versus catalytically active (blue) and dead (green) sACE2.v2.4-IgG1. $N = 8$ mice for the IgG1 group and catalytic active group, $N = 9$ for the catalytic dead group. Error bars for mouse weight are centered on the mean and show SEM. The Gehan–Breslow–Wilcoxon test was applied to analyze the survival curves. $P = 0.0099$ for the catalytic active group compared with the catalytic dead group. $P = 0.017$ for the catalytic dead group compared with IgG1 control.

D, E    Catalytically active sACE2.v2.4-IgG1 (D) and catalytically dead sACE2.v2.4(NN)-IgG1 (E) were immobilized on BLI biosensors and transferred to solutions of RBD (original variant) as the soluble analyte (0–60 s) and returned to buffer to measure dissociation (60–360 s). RBD concentrations are indicated on the right of the BLI sensorgrams.

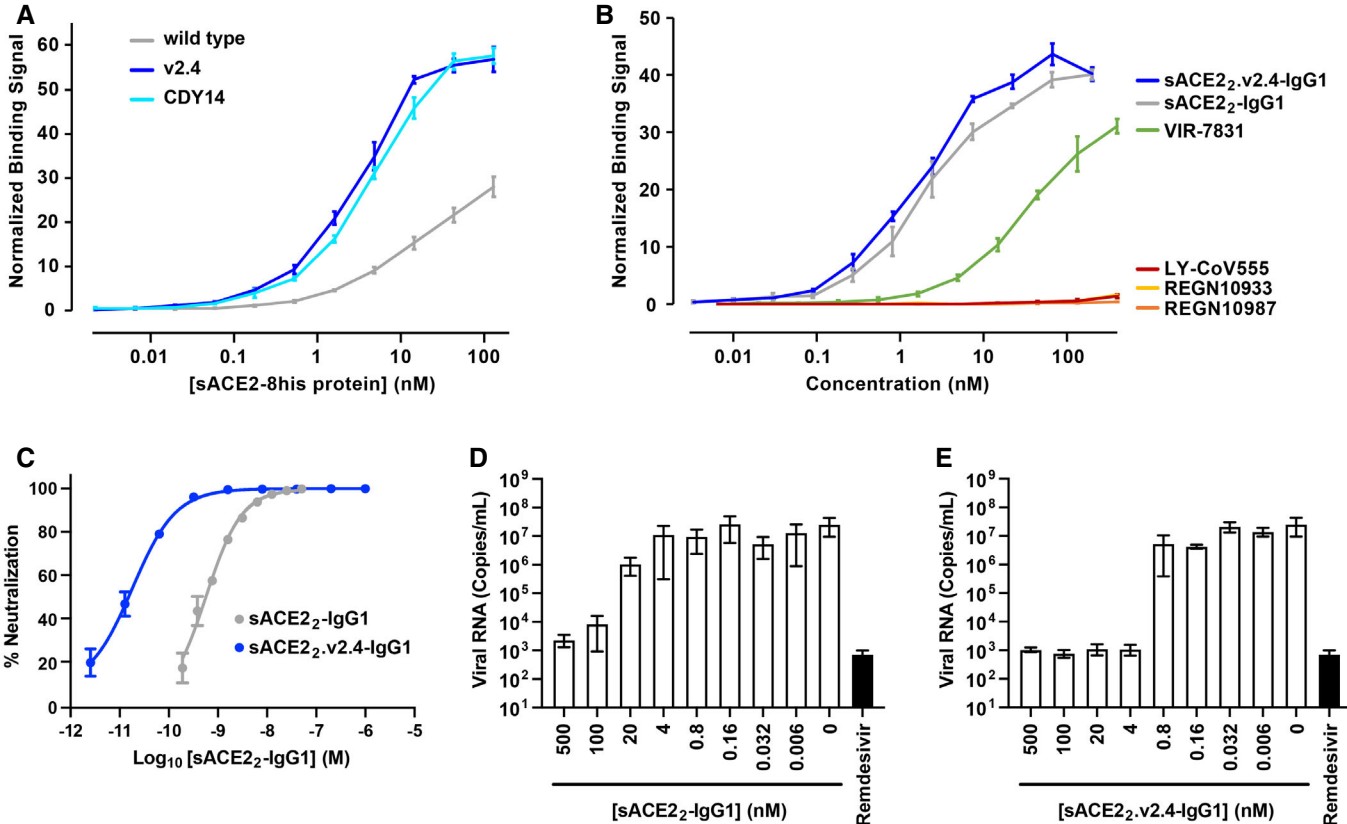

**Figure 4. The engineered decoy tightly binds S of BA.1 omicron and neutralizes infection.**

A  Binding of monomeric sACE2-8his proteins to Expi293F cells expressing S of BA.1 omicron was measured by flow cytometry. Data are mean ± SEM, N = 4 independent experiments.

B  Binding of dimeric sACE2₂-IgG1 proteins was compared with the binding of antibodies authorized for therapeutic use in COVID-19 patients. Binding to Expi293F cells expressing BA.1 omicron S was measured by flow cytometry. Data are mean ± SEM, N = 4 independent biological replicates.

C  Neutralization of BA.1 omicron pseudovirus. sACE2₂-IgG1 (gray) or sACE2₂.v2.4-IgG1 (blue) were incubated with pseudovirus for 1 h before adding to HeLa-hACE2-11 cells. Infection 48 h later was measured by luciferase reporter gene expression. Data are mean ± SD, N = 3 independent biological replicates.

D, E  Authentic BA.1 omicron virus (isolate USA/MD-HP20874/2021) was incubated with sACE2₂-IgG1 (D) or sACE2₂.v2.4-IgG1 (E) for 1 h and added to Calu-3 cells. Infection 48 h later was measured by RT–qPCR for the viral N gene. 3 μM remdesivir (black columns) is a positive neutralization control. Data are mean ± SD, N = 4 independent replicates.

T27Y, L79T, and N330Y), and a second engineered decoy called CDY14 (ACE2 mutations K31M, E35K, S47A, L79F, L91P, and N330Y; this is the highest affinity decoy reported in published literature (Sims *et al*, 2021)). Cells were washed and bound proteins were detected by flow cytometry. BA.1 omicron does not escape the engineered decoy receptors, and both sACE2.v2.4 and sACE2.CDY14 bind tighter than wild-type sACE2 (Fig 4A). We note that sACE2.CDY14, with twice as many mutations, did not bind any tighter than sACE2.v2.4.

The binding of dimeric sACE2₂.v2.4-IgG1 to cells expressing S of BA.1 omicron was compared with four monoclonal antibodies authorized for clinical use (Fig 4B). Whereas the decoy receptor bound to BA.1 omicron S at low nanomolar concentrations, no substantial binding was observed for REGN10933, REGN10987, or LY-CoV555. Of the tested antibodies, only VIR-7831 bound (consistent with prior reports (preprint: Ikemura *et al*, 2022; Planas *et al*, 2022; VanBlargan *et al*, 2022; Cao *et al*, 2022a)), albeit less tightly than the engineered decoy in this assay.

HeLa cells expressing human ACE2 were infected with a BA.1 omicron pseudovirus that contains a luciferase reporter gene.

Engineered sACE2₂.v2.4-IgG1 ($IC_{50}$ 18 ± 7 pM, based on the concentration of monomeric subunits) was over an order of magnitude more potent than wild-type sACE2₂-IgG1 ($IC_{50}$ 580 ± 70 pM) (Fig 4C), consistent with previous neutralization studies of other SARS-CoV-2 variants. We further tested the neutralization of authentic BA.1 omicron virus infecting Calu-3 cells. Based on quantitative measurements of viral RNA, we estimated the $IC_{50}$ for wild-type sACE2₂-IgG1 (Fig 4D) and engineered sACE2₂.v2.4-IgG1 (Fig 4E) to be 7.5 ± 9.2 nM and 0.14 ± 0.22 nM, respectively. We conclude that sACE2₂.v2.4-IgG1 remains exceptionally effective at neutralizing BA.1 omicron *in vitro*.

K18-hACE2 transgenic mice were inoculated with SARS-CoV-2 BA.1 omicron virus and treated with isotype-matched IgG1 control or sACE2₂.v2.4-IgG1 (15 mg/kg/day IV, once a day for 7 days, beginning 24 h postinoculation; Fig 5A). K18-hACE2 transgenic mice inoculated with SARS-CoV-2 BA.1 omicron at $1 \times 10^4$ PFU did not show any weight change (Fig EV1), which is consistent with the recent finding that SARS-CoV-2 BA.1 omicron causes mild disease in K18-hACE2 mice (Halfmann *et al*, 2022). Thus, we increased doses

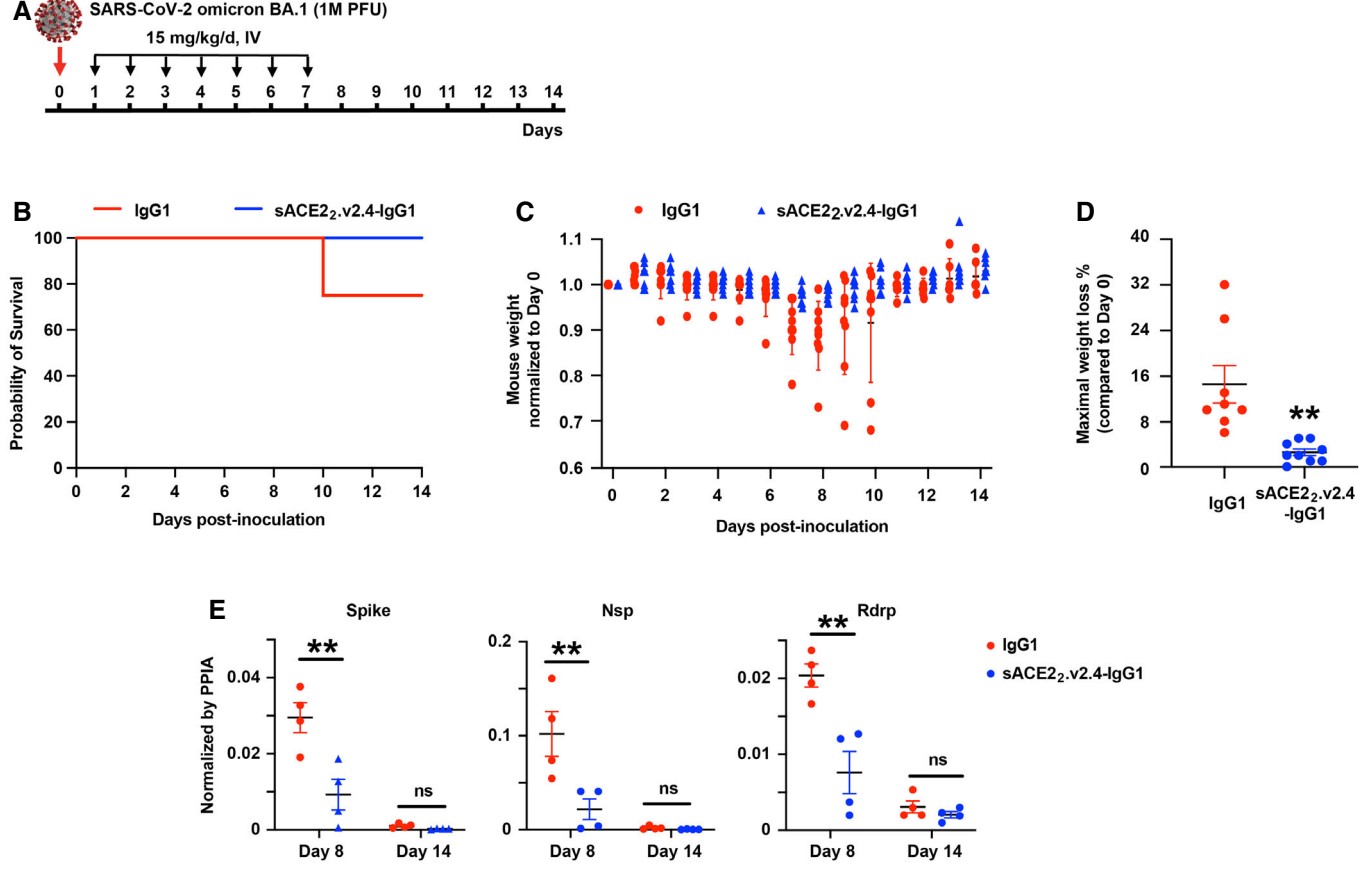

**Figure 5.** sACE2₂.v2.4-IgG1 inhibits SARS-CoV-2 BA.1 omicron replication *in vivo*.

A    K18-hACE2 transgenic mice were inoculated intranasally with $1 \times 10^6$ PFU SARS-CoV-2 BA.1 omicron (isolate USA/MD-HP20874/2021) and treated 24 h postinocu-
     lation with IgG1 isotype control or sACE2₂.v2.4-IgG1 (15 mg/kg/day intravenously, once per day for 7 days).

B, C  Survival (B) and weight loss (C). $N = 8$ (IgG1 control) or 9 (sACE2₂.v2.4-IgG1 treated) mice. Error bars for mouse weight are centered on the mean and show SEM.

D    Maximal weight loss of mice (compared with Day 0) within 14 days following BA.1 omicron virus infection. Data are presented as mean ± SEM, $N = 8$ (IgG1
     control) or 9 (sACE2₂.v2.4-IgG1 treated) mice. **$P < 0.01$ by two-ways ANOVA, corrected by Tukey.

E    Viral Spike, Nsp, and Rdrp mRNA levels in lung tissue were measured by RT–qPCR on Days 8 and 14, normalized to Ppia transcript levels. Data are presented as
     mean ± SEM, $N = 4$ mice per group. ns, not significant; **$P < 0.01$ by two-ways ANOVA, corrected by Tukey.

100-fold $(1 \times 10^6$ PFU) to induce weight loss. 25% of mice in the control group died on Day 10 and the maximal weight loss was up to 32% (Fig 5B–D). Mice treated with sACE2₂.v2.4-IgG1 survived to the experiment's conclusion on Day 14 and the maximal weight loss was up to 5% (Fig 5B–D). Levels of viral transcripts for Spike, Nsp, and Rdrp in lung tissue on Day 8 were significantly decreased in treated versus control mice, whereas viral transcript levels were low in both groups on Day 14 when the infection is resolved based on the recovery of body weight (Fig 5E).

**The engineered decoy binds tightly to S of BA.2 and BA.4/BA.5 omicron**

The S sequences of BA.1 and BA.2 are separated by 26 mutations and may therefore differ substantially in their interactions with binding proteins. BA.4 and BA.5 omicron variants have the same S sequences and differ from BA.2 at 5 positions. Using flow cytometry to measure the binding of monomeric sACE2(18–615) proteins to

BA.2 or BA.4/BA.5 S-expressing cells, it was observed that engineered decoys carrying the v2.4 or CDY14 mutations bound tighter than wild-type sACE2 (Fig 6A and D). Furthermore, avid binding of sACE2₂.v2.4-IgG1 to BA.2 or BA.4/BA.5 omicron S-expressing cells outperformed four monoclonal antibodies authorized for emergency use as therapeutics in nearly all cases (Fig 6B and E). In HeLa cells expressing human ACE2 that were infected with a BA.2 omicron pseudovirus carrying a luciferase reporter gene, sACE2₂.v2.4-IgG1 inhibited virus replication with an $IC_{50}$ of 130 ± 40 pM compared with 350 ± 90 pM for wild-type sACE2₂-IgG1 (Fig 6C).

**Molecular mechanism of affinity enhancement by the engineered decoy**

We previously used molecular dynamics simulations on the micro-to milli-second time scale to understand why ACE2-based decoys carrying the v2.4 mutations bind much tighter to S from the Wuhan, delta, and gamma variants (Zhang *et al*, 2022). Compared with

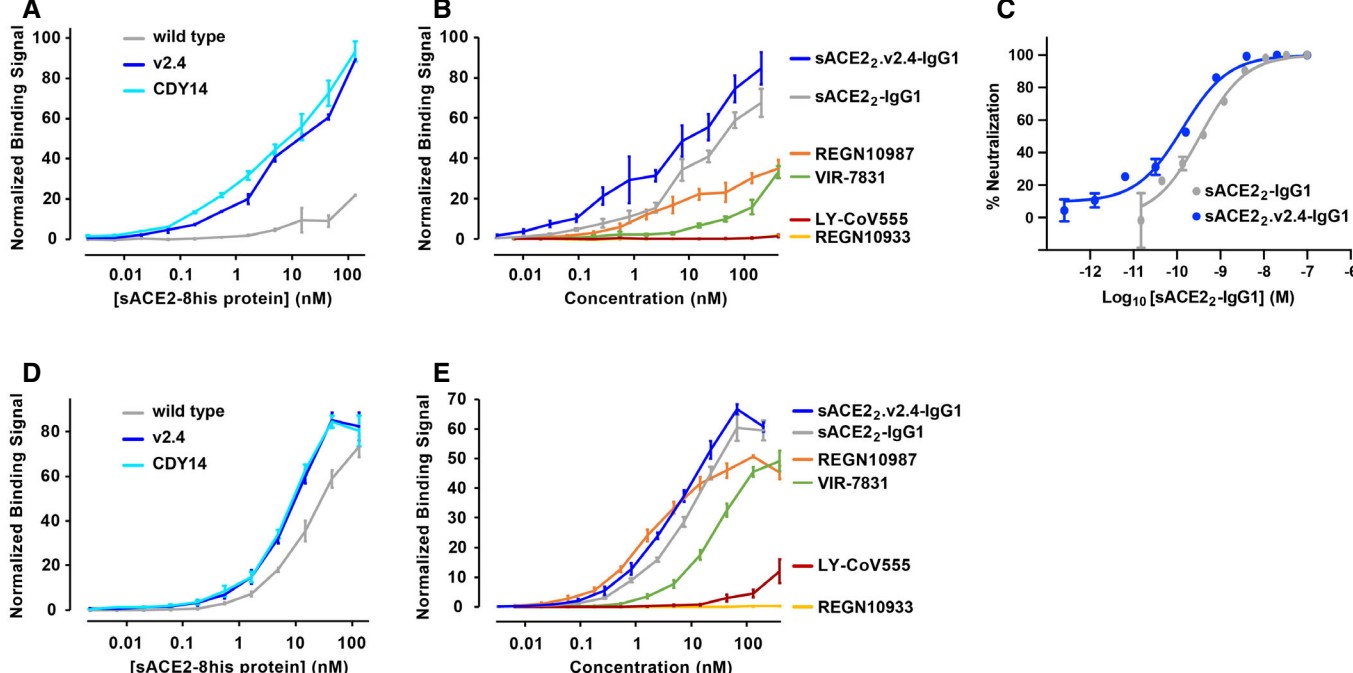

**Figure 6. The engineered decoy tightly binds S of BA.2 and BA.4/BA.5 omicron.**

A   Using flow cytometry, binding to Expi293F cells expressing S of BA.2 omicron was measured for monomeric sACE2-8his proteins. Data are mean ± SEM, $N$ = 3 independent biological replicates.

B   Binding of antibodies and dimeric sACE2$_2$-IgG1 proteins to Expi293F cells expressing BA.2 omicron S measured by flow cytometry. Data are mean ± SEM, $N$ = 4 independent experiments.

C   BA.2 omicron pseudovirus was incubated with sACE2$_2$-IgG1 (gray) or sACE2$_2$.v2.4-IgG1 (blue) for 1 h and added to HeLa-hACE2-11 cells. Infection was measured after 48 h. Data are mean ± SD, $N$ = 3 independent replicates.

D, E   Binding of monomeric sACE2-8his proteins (D) and dimeric IgG1 proteins (E) to Expi293F cells expressing BA.4/BA.5 omicron S as measured by flow cytometry. Data are mean ± SEM, $N$ = 3 independent experiments.

wild-type ACE2, ACE2.v2.4 was found to make new atomic contacts to the RBD and is "prestabilized" in the RBD-bound conformation. Here, to understand why soluble derivatives of ACE2.v2.4 continue to bind with higher affinity than wild-type ACE2 to the RBD of omicron variants, we modeled the interacting proteins using ROSETTA (Leman *et al*, 2020). The cryo-electron microscopy structure of BA.1 omicron RBD bound to ACE2 (PDB 7WPB, 2.79 Å resolution) was used as a template for modeling ACE2.v2.4 bound to BA.1, BA.2, and BA.4/BA.5 RBDs (Yin *et al*, 2022). Structures were relaxed using the ROSETTA energy function (see Methods) and PDB files for all models are provided as Source Data to Fig 7.

BA.1 and BA.2 omicron RBDs have identical residues at the interface except at position 496 (serine in BA.1 and glycine in BA.2), which is 7.6 Å from ACE2-D38 (Cα-Cα distance). Due to their close similarity at the interface, we describe here the BA.1 omicron models as representative of atomic interactions by ACE2.v2.4 to both BA.1 and BA.2 omicron. Substitution T27Y in ACE2.v2.4 brings the aromatic ring of tyrosine-27 into a cluster of hydrophobic residues on the omicron RBD formed by RBD-F456, Y473, A475, and Y489 (Fig 7A). This is associated with minor backbone movements of RBD loop 1 (a.a. 455-491) and a shift of RBD-Y473 to resolve a small steric clash with the larger ACE2-T27Y side chain. We speculate that the dynamic flexibility of RBD loops accommodates mutations on the ACE2 surface to resolve minor clashes. The two other v2.4

mutations, ACE2-L79T and ACE2-N330Y, are at the interface periphery. RBD-F486 makes contacts to ACE2-L79T (Fig 7B), while RBD-T497 of RBD loop 2 (a.a. 496-506) moves closer to pack against ACE2-N330Y (Fig 7C). ACE2.v2.4 forms new polar contacts between the side chain hydroxyls of ACE2-T27Y and RBD-Y473 and between the hydroxyl of ACE2-N330Y and backbone carbonyl of RBD-P499 that are not seen in the complexes with wild-type ACE2 (compare the wild type/gray and ACE2.v2.4/colored structures in Fig 7A and 7C). The new atomic contacts formed, together with increased hydrophobic packing around ACE2-T27Y, may all contribute to the enhanced affinity of ACE2.v2.4 and were observed and described in our prior molecular dynamics-based modeling of ACE2.v2.4 bound to the RBDs of SARS-CoV-2 Wuhan, delta, and gamma variants (Zhang *et al*, 2022), suggesting that the atomic mechanisms of enhanced affinity are shared across all the virus variants.

BA.4/BA.5 omicron S differs from the other omicron sublineages at two sites in the interface with ACE2. First, there is glutamine at RBD position 493 (BA.1 and BA.2 both have arginine); this reverts position 493 back to the amino acid found in the original SARS-CoV-2 S sequence against which ACE2.v2.4 was engineered for enhanced affinity. Second, RBD-F486 is mutated to valine and is modeled to make direct atomic contacts with ACE2-L79 in wild-type ACE2 (RBD-486V Cγ2 is 3.8 Å from ACE2-L79 Cδ2) or with ACE2-L79T in the engineered ACE2.v2.4 decoy (RBD-486V Cγ1 and Cγ2 are 3.9 Å

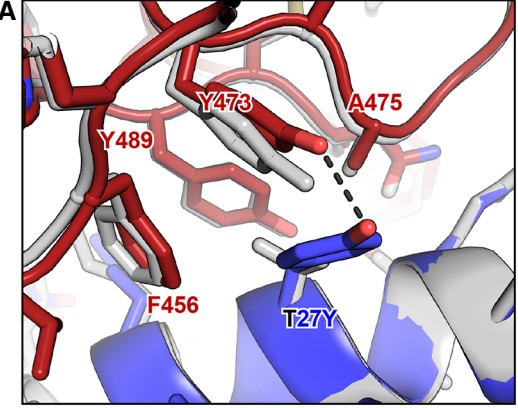
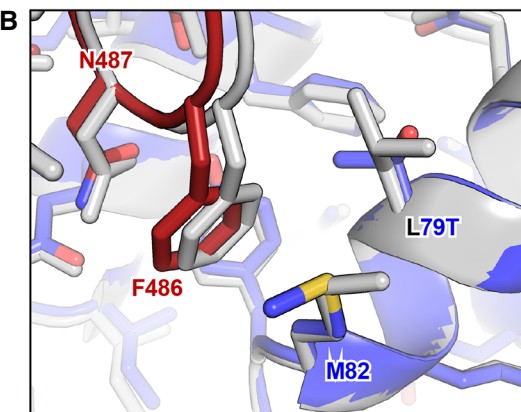
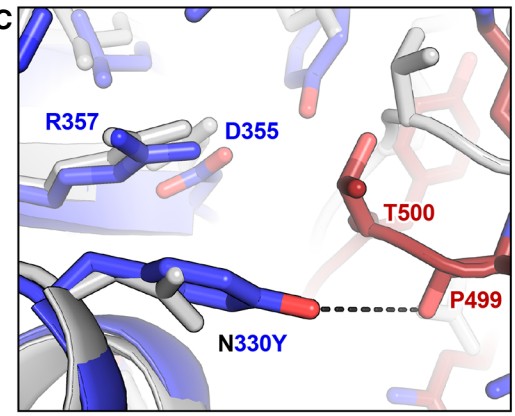
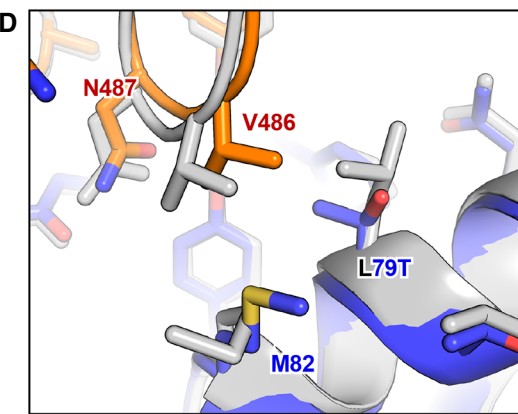

**Figure 7. Molecular basis for the enhanced affinity of the engineered decoy for omicron RBD.**

A–C The BA.1 omicron RBD (dark red for carbon atoms, CPK coloring for noncarbon atoms) bound to wild-type ACE2 (gray) and engineered ACE2.v2.4 (blue for carbon atoms, CPK coloring for noncarbon atoms) was modeled using ROSETTA. Superpositions of the models are shown in the regions surrounding ACE2.v2.4 mutations T27Y (A), L79T (B), and N330Y (C). New polar contacts formed by the ACE2.v2.4 mutations are indicated with dashed black lines.

D Model of BA.4/BA.5 omicron RBD (orange for carbon atoms, CPK coloring for noncarbon atoms) bound to wild-type ACE2 (gray) and engineered ACE2.v2.4 (blue for carbon atoms, CPK coloring for noncarbon atoms). The region surrounding ACE2.v2.4 mutation L79T is shown.

PDB files of models are provided as Source Data.

Source data are available online for this figure.

and 3.8 Å from ACE2-79T Cγ2) (Fig 7D). Consequently, modeling predicts that the RBD-F486V substitution interacts favorably with both wild-type receptors and the engineered decoy, consistent with sACE2.v2.4-IgG1 maintaining a high affinity to BA.4/BA.5 omicron (Fig 6E).

# Discussion

The future of the SARS-CoV-2 pandemic is uncertain, but based on the history of the past 2 years, it is expected that new virus variants will continue to emerge as SARS-CoV-2 becomes endemic. There will likely be a continuing need for effective therapeutics, especially as immunity wanes, vaccine hesitancy remains high, and new virus variants emerge that escape natural and vaccine-induced antibodies.

Monoclonal antibodies have been important drugs in the clinic and can be co-administered with small molecule drugs that target other features of the SARS-CoV-2 replication cycle. Alarmingly,

omicron variants have accumulated mutations to partially or fully escape many anti-S antibodies. It is therefore unclear whether the continuous development of new monoclonal antibodies is a feasible long-term strategy as new variants continue to emerge. We show here that decoy receptors remain highly potent against omicron variants and based on their similarity to the native ACE2 receptor, decoys will likely remain effective against future variants as SARS-CoV-2 evolves.

We also addressed the important question of whether sACE2 has additional therapeutic benefits in a SARS-CoV-2 infection beyond the direct binding of the viral spike protein. There has been disagreement in the literature as to whether sACE2-catalyzed turnover of vasoconstrictive and pro-inflammatory peptides will confer therapeutic benefits or whether it is a safety liability. Many groups knocked out the catalytic activity when developing candidate decoy receptors, negating ill-defined risks of adverse hypotension (preprint: Cohen-Dvashi *et al*, 2020; Glasgow *et al*, 2020; preprint: Iwanaga *et al*, 2020; Lei *et al*, 2020; preprint: Chen *et al*, 2021;

Higuchi *et al*, 2021; Sims *et al*, 2021; Tanaka *et al*, 2021). However, we show here that catalytically inactivated sACE2$_2$.v2.4(NN)-IgG1 is not as effective at prolonging the survival of hACE2 transgenic mice infected with a lethal virus dose, suggesting that the catalytic activity of ACE2 present in the decoy confers additional therapeutic benefits. We speculate that while neutralizing antibodies are most effective when administered early to patients with mild-to-moderate disease, decoy receptors may have broader reach into hospitalized patient groups due to both neutralizing and ACE2 catalytic activities.

We previously tested intravenously administered sACE2$_2$.v2.4-IgG1 in prophylactic and therapeutic regimens in K18-hACE2 mice, finding that 50-100% of mice survived lethal doses of original and gamma viruses (Zhang *et al*, 2022). We now show the protein effectively delays death when inhaled, a mode of delivery that has significant clinical relevance. It is possible that even greater efficacy may be achieved by increasing the dose amount and frequency. Inhalation can be readily administered in an outpatient setting and would help reduce the need for in-hospital treatment, which is especially important when hospital resources become scarce during COVID-19 "surges." Inhalation may be a first-line treatment in outpatients with early infection, whereas intravenous delivery could be reserved for hospitalized patients in which the virus has spread beyond the lungs.

The studies here strengthen the concept of ACE2-based decoy receptors as broadly effective neutralizing agents for SARS-CoV-2 variants with multiple therapeutic mechanisms. Next-generation sACE2 decoys with enhanced S affinity and neutralization potency are promising drug candidates for treating an ever-evolving threat long into the future.

# Materials and Methods

### Cell lines

Expi293F cells (Thermo Fisher) were cultured in Expi293 Expression Medium (Thermo Fisher), 37°C, 125 r.p.m., 8% CO$_2$. HeLa-hACE2-11 (a stable human ACE2 HeLa clone) were grown in Dulbecco's Modified Eagle's Medium (DMEM) high glucose (4500 mg/l) with 10% fetal bovine serum (FBS), 100 units/ml penicillin, and 100 μg/ml streptomycin at 37°C, 5% CO$_2$. Calu-3 (ATCC HTB-55) cells were grown in Modified Eagle's Medium high glucose (4500 mg/l) with 10% FBS, 4 mM L-Glutamine, 1 mM sodium pyruvate, 100 units/ml penicillin, and 100 μg/ml streptomycin at 37°C with 5% CO$_2$. ExpiCHO-S cells (Thermo Fisher) were cultured in ExpiCHO Expression Medium (Thermo Fisher) at 37°C, 125 r.p.m., 8% CO$_2$. Vero E6 (CRL-1586, American Type Culture Collection) were cultured at 37°C, 5% CO$_2$, in DMEM supplemented with 10% FBS, 1 mM sodium pyruvate, 1× nonessential amino acids, 100 units/ml penicillin, and 100 μg/ml streptomycin. Cell lines supplied by commercial vendors were not further authenticated.

### Expression of proteins

All genes were cloned into the NheI-XhoI sites of pcDNA3.1(+) (Invitrogen) with a consensus Kozak sequence (GCCACC) upstream of the start ATG. Plasmids for sACE2$_2$-IgG1 (Addgene #154104),

sACE2$_2$.v2.4-IgG1 (#154106), sACE2(18-615)-8his (#149268), sACE2 (18-615).v2.4-8his (#149664), and Wuhan RBD-8his (#145145) are available from Addgene. Mutations for the CDY14 decoy receptor were introduced into the wild-type sACE2 plasmids using extension overlap PCR and confirmed by Sanger sequencing.

All soluble proteins (with the exception of proteins used in the aerosolization and inhalation experiment) were expressed in Expi293F cells transfected using Expifectamine (Thermo Fisher) according to the manufacturer's instructions. Transfection enhancers 1 (5 μl per ml of culture) and 2 (50 μl per ml of culture) were added ~18 h post-transfection, and the medium was collected on Day 6.

For the aerosolization and inhalation experiment, large-scale production of sACE2$_2$-IgG1 proteins was in ExpiCHO-S transfected with plasmids (1,000 ng per ml of culture) using ExpiFectamine CHO (Thermo Fisher) according to the manufacturer's instructions. ExpiFectamine CHO Enhancer (6 μl per ml of culture) was added ~20 h post-transfection and the temperature was decreased to 33°C. On Days 1 and 5, ExpiCHO feed (240 μl per ml of culture) was added. CO$_2$ was decreased over Days 9–12 to 5%, and the medium was collected on Days 12-14.

### Purification of sACE2$_2$-IgG1 and monoclonal antibodies

Expression medium was collected after removal of cells by centrifugation (800 *g*, 4°C, 10 min) and the pH was adjusted to 7.5 by adding 1 M Tris base. The medium was centrifuged (15,000 *g*, 4°C, 20 min) and incubated for 2 h at 4°C with 2 ml KanCapA resin (Kaneka Corporation) per 100 ml. The resin was collected in a chromatography column, washed with 10 column volumes (CV) of Dulbecco's phosphate-buffered saline (PBS), and protein eluted with 4 CV 60 mM sodium acetate (pH 3.7) into 2 CV 1 M Tris (pH 8.0). The pH was raised to 7–8 by 1–2 CV 1 M Tris base. Eluates were concentrated by centrifugal filtration and separated by size exclusion chromatography using PBS as the running buffer. Peak fractions were pooled, concentrated, and aliquots were stored at −80°C after snap freezing in liquid nitrogen. Concentrations were determined by absorbance at 280 nm using calculated molar extinction coefficients. For consistency, all concentrations in this manuscript are based on monomeric sACE2 subunits (sACE2$_2$.v2.4-IgG1 has a monomer molecular weight of 108.4 kD) or an H + L chain for antibodies (i.e., concentrations can be considered a measure of binding sites).

The isotype-matched control antibody was from Bio X Cell (InVivoPlus human IgG1 isotype control, Cat. No. BP0297).

### Purification of sACE2-8his and RBD-8his

The expression medium was centrifuged twice (800 *g*, 4°C, 10 min, followed by 15,000 g, 4°C, 20 min) and supernatants were incubated at 4°C, 90 min, with 1 ml HisPur Ni-NTA Resin (Thermo Fisher) per 100 ml. The resin was collected in a chromatography column, washed with >20 CV PBS, washed with ~10 CV PBS containing 20 mM imidazole, and proteins eluted with ~15 CV PBS containing 250 mM imidazole (pH 8.0). Eluates were concentrated by centrifugal filtration and proteins were separated on a Superdex 200 Increase 10/300 GL column (GE Healthcare) with running buffer PBS. Peak fractions at the expected molecular weight were pooled,

concentrated, and aliquots were stored at −80°C after snap freezing in liquid nitrogen. Concentrations were based on absorbance at 280 nm using calculated molar extinction coefficients.

## S binding assay

Human codon-optimized genes encoding N-terminal myc-tagged S proteins of BA.1, BA.2, and BA.4/BA.5 omicron were synthesized (Integrated DNA Technologies) and cloned into the NheI-XhoI sites of pcDNA3.1(+) (Invitrogen). The S sequences used in this manuscript have the following mutations from the Wuhan reference sequence (GenBank Accession No. YP_009724390). BA.1: A67V, H69del, V70del, T95I, G142D, V143del, V144del, Y145del, N211del, L212I, ins214EPE, G339D, S371L, S373P, S375F, K417N, N440K, G446S, S477N, T478K, E484A, Q493R, G496S, Q498R, N501Y, Y505H, T547K, D614G, H655Y, N679K, P681H, N764K, D796Y, N856K, Q954H, N969K, and L981F. BA.2: T19I, L24del, P25del, P26del, A27S, G142D, V213G, G339D, S371F, S373P, S375F, T376A, D405N, R408S, K417N, N440K, S477N, T478K, E484A, Q493R, Q498R, N501Y, Y505H, D614G, H655Y, N679K, P681H, N764K, D796Y, L849P, Q954H, and N969K. BA.4/BA.5: T19I, L24del, P25del, P26del, A27S, H69del, V70del, G142D, V213G, G339D, S371F, S373P, S375F, T376A, D405N, R408S, K417N, N440K, L452R, S477N, T478K, E484A, F486V, Q498R, N501Y, Y505H, D614G, H655Y, N679K, P681H, N764K, D796Y, Q954H, and N969K. Expi293F cells were transfected with 500 ng plasmid DNA per ml of culture using Expifectamine (Thermo Fisher) according to the manufacturer's directions. After 24-28 h, cells were washed with cold PBS containing 0.2% bovine serum albumin (PBS-BSA) and added to 3-fold serial titrations of the binding proteins in 96-well round-bottomed plates. Plates were incubated on ice for 30 min with regular agitation. Cells were washed with PBS-BSA. For assays examining the binding of monomeric sACE2(18-615)-8his, cells were resuspended in PBS-BSA containing 1:150 polyclonal chicken anti-HIS-FITC (Immunology Consultants Laboratory) and 1:300 anti-myc-Alexa 647 (clone 9B11, Cell Signaling Technology). For assays examining the avid binding of dimeric sACE2$_2$-IgG1 and monoclonal antibodies, cells were resuspended in PBS-BSA containing 1:150 polyclonal chicken anti-MYC-FITC (Immunology Consultants Laboratory) and 1:300 anti-human IgG-APC (clone HP6017, BioLegend). Plates were incubated for 30 min on ice with occasional mixing, washed twice with PBS-BSA, and analyzed on a BD Accuri C6 flow cytometer using CFlow version 1.0.264.15. Cells were gated by forward and side scatter to exclude dead cells and debris. Cells expressing Spike were then gated based on detection of the N-terminal myc tag and the binding of sACE2 or monoclonal antibodies was measured for this myc-positive population. No binding of sACE2 or monoclonal antibodies is observed in the myc-negative population. Binding data are presented as the mean fluorescence units (FITC for bound 8his proteins and APC for bound IgG1 proteins) with subtraction of background fluorescence from cells incubated without sACE2 proteins. Data were normalized across independent experiments based on the total measured fluorescence in each experiment.

## Biolayer interferometry (BLI)

sACE2$_2$.v2.4-IgG1 and sACE2$_2$.v2.4(NN)-IgG1 were diluted in assay buffer (10 mM HEPES pH 7.6, 150 mM NaCl, 3 mM EDTA, 0.05%

polysorbate 20, and 0.5% nonfat dry milk) to 100 nM and immobilized for 60 s to anti-human IgG Fc capture biosensors (Sartorius). Sensors were transferred to assay buffer for 30 s to set the baseline, then transferred to Wuhan RBD-8his for 60 s (association) and transferred back to buffer for 300 s (dissociation). Data were collected on an Octet RED96a and analyzed using instrument software (Molecular Devices) with a global fit 1:1 binding model.

## Catalytic activity assay

ACE2 activity was measured with the Fluorometric ACE2 Activity Assay Kit (BioVision) according to the manufacturer's directions. Fluorescence was read on a Biotek Cytation 5 instrument.

## SARS-CoV-2 amplification and quantification

All experiments with live SARS-CoV-2 were performed in a BSL3 level laboratory and with approval from the office of Environmental Health and Safety at the University of Illinois at Chicago prior to the initiation. SARS-CoV-2 isolate hCoV-19/Japan/TY7-503/2021 (P.1/gamma), SARS-CoV-2 isolate 2019n-CoV/USA-WA1/2020 (original variant), and SARS-CoV-2 isolate USA/MD-HP20874/2021 (omicron BA.1) were obtained from BEI Resources (# NR-54982, NR-52281, and NR-56461), NIAID, NIH. P.1/gamma and WA1/2020 viruses were propagated in Vero E6 cells. The omicron BA.1 variant was propagated in Calu-3 cells. Culture supernatants were collected upon observation of cytopathic effects. Cell debris was removed by centrifugation and passing through a 0.22 μm filter. The supernatant was aliquoted and stored at −80°C. Virus titers were quantitated by a plaque-forming assay using Vero E6 cells.

## Inoculation of K18-hACE2 transgenic mice with SARS-CoV-2 variants

Biosafety level 3 (BSL-3) protocols for animal experiments with live SARS-CoV-2 were performed by personnel equipped with powered air-purifying respirators in strict compliance with NH guidelines for humane treatment and approved by the University of Illinois Animal Care & Use Committee (ACC protocol 21-055 and IBC protocol 20-036). Hemizygous K18-hACE2 mice (strain 034860: B6.Cg-Tg(K18-ACE2)2Prlmn/J) were purchased from The Jackson Laboratory. Animals were housed in groups and fed standard chow. Mice (10-16 weeks old) were anesthetized by ketamine/xylazine (50/5 mg/kg, IP). Mice were then inoculated intranasally with $1 \times 10^4$ PFU of SARS-CoV-2 gamma or WA-1/2020 variants, or $1 \times 10^6$ PFU SARS-CoV-2 BA.1 omicron variant, suspended in 20 μl of sterile PBS. Animals with sex- and age-matched littermates were randomly included in experiments. No animals were excluded attributed to illness after experiments. Animal experiments were carried out in a blinded fashion whenever feasible.

## Administration of sACE2$_2$.v2.4-IgG1 by inhalation

Mice were placed in a pie cage for aerosol delivery (Braintree Scientific, # MPC-3-AERO). The mice were individually separated, and the pie cage can hold as many as 11 mice. An MPC Aerosol Medication Nebulizer (Braintree Scientific, # NEB-MED H) aerosolized 7.5 ml sACE2$_2$.v2.4-IgG1 (8.3 mg/ml in PBS) in the nebulizer

cup and delivered the aerosol to the mice in the pie cage. Inhalation delivery took approximately 25 min. sACE2$_2$.v2.4-IgG1 was administered 3 times with doses 36 h apart and starting 12 h post virus inoculation. PBS was delivered as control. For the number of animals needed to achieve statistically significant results, we conducted an *a priori* power analysis. We calculated power and sample sizes according to data from pilot experiments, variations within each group of data, and variance similarities between the groups were statistically compared. Animals with sex- and age-matched littermates were randomly included in experiments. No animals were excluded attributed to illness after experiments. Animal experiments were carried out in a blinded fashion whenever feasible.

### mRNA expression measured by quantitative RT–qPCR

Tissues were homogenized in 1 ml Trizol solution (Thermo Fisher, # 15596026). Tissue homogenates were clarified by centrifugation at 10,000 rpm for 5 min and stored at −80°C. RNA was extracted according to the Trizol protocol. RNA was quantified by Nanodrop 1000 (Thermo Fisher) and reverse transcribed with Superscript III (Invitrogen # 18080093) using random primers. FastStart Universal SYBR Green Master Mix (Thermo Fisher # 4913850001) was used for the relative quantification of cDNA on the ViiA 7 Real-Time PCR System (Thermo Fisher). Primer information is included in Table 2.

### Histology and imaging

Animals were euthanized before harvesting and fixation of tissues. Lung lobes were fixed with 4% PFA (paraformaldehyde) for 48 h before further processing. Tissues were embedded in paraffin and sections were stained with hematoxylin and eosin. The slides were

Table 2. qPCR primer sequences and product size.

| Target Gene | Primer Sequences | | Size (bp) |
|---|---|---|---|
| SARS-CoV-2 Spike | Forward | GCTGGTGCTGCAGCTTATTA | 107 |
| | Reverse | AGGGTCAAGTGCACAGTCTA | |
| SARS-CoV-2 Nsp | Forward | CAATGCTGCAATCGTGCTAC | 117 |
| | Reverse | GTTGCGACTACGTGATGAGG | |
| SARS-CoV-2 Rdrp | Forward | AGAATAGAGCTCGCACCGTA | 101 |
| | Reverse | CTCCTCTAGTGGCGGCTATT | |
| Tnf | Forward | ACGGCATGGATCTCAAAGAC | 138 |
| | Reverse | AGATAGCAAATCGGCTGACG | |
| Ifng | Forward | ACAATGAACGCTACACACTGCAT | 71 |
| | Reverse | TGGCAGTAACAGCCAGAAACA | |
| Il1a | Forward | TTGGTTAAATGACCTGCAACA | 122 |
| | Reverse | GAGCGCTCACGAACAGTTG | |
| Il1b | Forward | GCAACTGTTCCTGAACTCAACT | 89 |
| | Reverse | ATCTTTTGGGGTCCGTCAACT | |
| Ppia | Forward | CAGTGCTCAGAGCTCGAAAGTTT | 66 |
| | Reverse | TCTCCTTCGAGCTGTTTGCA | |

scanned by Aperio Brightfield 20x. Images were taken by Aperio ImageScope 12.4.3 and analyzed by Zen software (Zeiss).

### Pseudovirion production

Pseudoviruses were created using plasmids for SARS-CoV-2 Omicron B.1.1.529 BA.1 Spike, SARS-CoV-2 Omicron B.1.1.529 BA.2 Spike (pcDNA3.3_SARS2_omicron BA.2 was a gift from David Nemazee, Addgene plasmid # 183700), and HIV-1 proviral vector pNL4-3.Luc.R⁻E⁻ (from the NIH AIDS Research and Reference Reagent Program) containing a luciferase reporter gene. Pseudovirions were created following a polyethylenimine (PEI)-based transient co-transfection on 293T cells. After 5 h, cells were washed with PBS and the medium was replaced with phenol red-free DMEM. 16 h post-transfection, supernatants were collected and filtered through a 0.45-μm pore size filter.

### Pseudovirus inhibition assay

HeLa-hACE2-11 cells were seeded ($5x10^3$ cells/well) onto white-bottomed 96-well tissue culture plates (100 μl/well) and incubated for 16 h, 37°C, 5% CO$_2$. The decoy sACE2 was titrated in pseudovirus supernatant and incubated at room temperature for 1 h. The pseudovirus/sACE2 mixtures were added to the target cells. Plates were incubated for 48 h and the degree of viral entry was determined by luminescence using the neolite reporter gene assay system (PerkinElmer). IC$_{50}$ values were determined by fitting dose–response curves with four-parameter logistic regression in GraphPad Prism 8 software.

### Live omicron virus neutralization assay

48 h prior to treatment, $3 \times 10^5$ Calu-3 cells/well were seeded into 24-well plates. Infection was with a clinical isolate of the SARS-CoV-2 omicron BA.1 variant (isolate USA/MD-HP20874/2021) from BEI Resources (NR-56461). Nontreatment controls, 5-fold serial dilutions of decoy sACE2 (final concentrations 500 nM – 0.006 nM), and a high concentration of positive control remdesivir (3 μM) were added to the same volume of SARS-CoV-2 (final MOI = 0.01) and incubated at room temperature for 1 h. Then, the mixture was added to the monolayer of cells and incubated for 1 h at 37°C, with 5% CO$_2$. The mixture was removed, cells washed with PBS, and monolayers overlaid with infection media (2% FBS). After 48 h, 100 μl of cell supernatants were collected and added to 300 μl of TRIzol. RNA was isolated using Invitrogen's PureLink RNA Mini Kits according to the manufacturer's protocol. Quantitative RT–qPCR was carried out using 5 μl of RNA template in TaqMan Fast Virus 1-Step Master Mix using primers and probes for the N gene (N1 primers) designed by the U.S. Centers for Disease Control and Prevention (IDT cat# 10006713). A standard curve was generated using dilutions of synthetic RNA from the SARS-CoV-2 amplicon region (BEI Resources, NR-52358). All experiments prior to RNA isolation were performed in a Biosafety Level 3 facility.

### Structural modeling

Models of SARS-CoV-2 omicron variants BA.1, BA.2, and BA.4/BA.5 RBDs bound to ACE2 and ACE2.v2.4 were generated using template

PDB 7WPB. 7WPB captures the structure of BA.1 RBD complexed with human ACE2. Models were generated using one round of ROSETTA Relax Design protocol with ROSETTA score function ref2015 and an appropriate mutation set defined for each complex. Relax Design allows for backbone perturbations for the entire structure, making it ideal for allowing movement beyond mutation sites to better accommodate desired mutants (Conway et al, 2014). Following Relax Design, each complex underwent an additional 5 rounds of ROSETTA Relax with score function ref2015. Lowest energy models were selected from the resulting 5 decoys per complex.

### Statistics and reproducibility

Quantification of replicate experiments is presented as mean ± SD or SEM as described in figure legends. Statistical tests are described in figure legends. Animals with sex- and age-matched littermates were randomly included in experiments. No animals were excluded attributed to illness after experiments. Animal experiments were carried out in a blinded fashion whenever feasible. Based on our experience, we expect changes in the gene/protein expression and function measurements to be detected with 4 mice per group, so the effect size was determined as $N = 4$ independent mice. The variance between groups that are being statistically compared was similar.

### Study approval

All aspects of this study were approved by the office of Environmental Health and Safety at the University of Illinois at Chicago prior to the initiation of this study. All work with live SARS-CoV-2 was performed in a BSL-3 laboratory by personnel equipped with powered air-purifying respirators.

## Data availability

This study includes no data deposited in external repositories. Structural models are provided in PDB format as Source Data in Fig 7.

**Expanded View** for this article is available online.

### Acknowledgements

This work was supported in part by NIH grants R43-AI162329 to EP and KKC; R01HL157489 to LZ; R01-HL162308 to ABM and JR. The following reagents were obtained through BEI Resources: isolate hCoV-19/Japan/TY7-503/2021 (P.1; NR-54982), isolate 2019n-CoV/USA-WA1/2020 (original; NR-52281), and isolate hCoV-19/USA/MD-HP20874/2021 (BA.1; NR-56461). Flow cytometry services were used at the Roy J. Carver Biotechnology Center, University of Illinois.

### Author contributions

**Lianghui Zhang:** Conceptualization; formal analysis; investigation; methodology; writing—original draft; writing—review and editing. **Krishna K Narayanan:** Formal analysis; investigation; methodology; writing—review and editing. **Laura Cooper:** Formal analysis; investigation; methodology; writing—review and editing. **Kui K Chan:** Formal analysis; funding acquisition; investigation; writing—review and editing. **Savanna S Skeeters:** Investigation; writing—review and editing. **Christine A Devlin:** Formal analysis; investigation; methodology; writing—review and editing. **Aaron Aguhob:** Investigation; methodology; writing—review and editing. **Kristie Shirley:** Investigation. **Lijun Rong:** Supervision; investigation; methodology; project administration; writing—review and editing. **Jalees Rehman:** Conceptualization; supervision; funding acquisition; project administration; writing—review and editing. **Asrar B Malik:** Conceptualization; supervision; funding acquisition; project administration; writing—review and editing. **Erik Procko:** Conceptualization; formal analysis; supervision; funding acquisition; investigation; methodology; writing—original draft; project administration; writing—review and editing.

In addition to the CRediT author contributions listed above, the contributions in detail are:
LZ performed the mouse studies. KKN cloned omicron plasmids, expressed proteins, and tested binding. LMC and LR tested pseudovirus and authentic virus neutralization. KKC measured BLI kinetics and purified proteins. SSS purified proteins. CAD examined ACE2 catalytic activity. AA modeled structures. KS cloned plasmids and purified proteins. EP purified proteins. EP and LZ drafted the manuscript. LZ, LR, JR, ABM, and EP supervised research and planned experiments. All authors contributed to the manuscript edits.

### Disclosure statement and competing interests

EP, ABM, LZ, and JR are co-inventors of a patent filing by the University of Illinois covering engineered decoy receptors that are licensed to Cyrus Biotechnology. Employees of Cyrus Biotechnology receive stock options.

### For more information

i https://www.nytimes.com/interactive/2020/science/coronavirus-drugs-treatments.html
ii https://www.fda.gov/emergency-preparedness-and-response/mcm-legal-regulatory-and-policy-framework/emergency-use-authorization#coviddrugs

---

**The paper explained**

**Problem**
SARS-CoV-2 variants continue to emerge, which escape antibodies administered in the clinic or antibodies that are elicited by vaccines and natural infection. This is especially true of omicron sublineages, which have accumulated significant mutations in the Spike proteins that decorate the virus surface.

**Results**
We have observed that an engineered soluble decoy receptor, based on the endogenous virus receptor on human cells (an enzyme called ACE2), tightly binds to the Spike proteins of a wide range of SARS-CoV-2 variants. When delivered by inhalation, this soluble decoy is therapeutically effective in a mouse model of COVID-19, and the enzymatic activity of the decoy also contributes to protecting against death. Importantly, the engineered decoy blocks the Spike proteins of BA.1, BA.2, and BA.4/BA.5 omicron sublineages.

**Impact**
Inhalation of a decoy drug is more suitable for outpatients early on during SARS-CoV-2 infection to neutralize virus replication, whereas intravenous administration may be indicated at later stages of disease when virus replication has spread and requires systemic delivery. While many monoclonal antibody drugs have lost potency against newer SARS-CoV-2 variants, decoy receptors remain relevant despite extensive changes to the virus Spike.

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
