## [Review Process File · EMBO Molecular Medicine]

An ACE2 decoy can be administered by inhalation and potently targets omicron variants of SARS-CoV-2

Lianghui Zhang, Krishna Narayanan, Laura Cooper, Kui Chan, Savanna Skeeters, Christine Devlin, Aaron Aguhob, Kristie Shirley, Lijun Rong, Jalees Rehman, Asrar Malik, and Erik Procko

DOI: [10.15252/emmm.202216109](https://doi.org/10.15252/emmm.202216109)

Corresponding authors: Erik Procko (procko@illinois.edu), Jalees Rehman (jalees@uic.edu), Lijun Rong (lijun@uic.edu), Asrar Malik (abmalik@uic.edu), Lianghui Zhang (lh Zhang@uic.edu)

Review Timeline:

Submission Date:	31st Mar 22
Editorial Decision:	9th May 22
Revision Received:	4th Aug 22
Editorial Decision:	30th Aug 22
Revision Received:	7th Sep 22
Accepted:	9th Sep 22

Editor: *Zeljko Durdevic*

Transaction Report:

9th May 2022

Dear Dr. Procko,

Thank you for the submission of your manuscript to EMBO Molecular Medicine. We have now received feedback from the three reviewers who agreed to evaluate your manuscript. As you will see from the reports below, the referees acknowledge the interest of the study but also raise important concerns that should be addressed in a major revision.

We would welcome the submission of a revised version within three months for further consideration. Please let us know if you require longer to complete the revision.

Please use this link to login to the manuscript system and submit your revision: in ot aila le

I look forward to receiving your revised manuscript.

Yours sincerely,

Zeljko Durdevic

**** Reviewer's comments ****

Referee #1 (Comments on Novelty/Model System for Author):

The authors should use the proper antibody control for animal experiment.

Referee #1 (Remarks for Author):

The authors had previously shown that The high-affinity and catalytically active decoy sACE2.v2.4-IgG1 was effective in vivo against SARS-CoV-2 variants when administered intravenously. In this study, the authors describe the inhalation of sACE2.v2.4-IgG1 could increase survival and ameliorate lung injury in K18-hACE2 transgenic mice inoculated with a lethal dose of the virulent P.1/gamma virus of decoy ACE2 receptor. The authors also confirm the catalytic activity of ACE2s confers the additional therapeutic benefit. Finally, they proved that ACE2s could tightly bind to omicron BA.1 and BA.2 in vitro binding assay. While the manuscript seems to be ready for publish, but there are some important issues need to be address.

1. The recent articles (Leyun Wu, Signal Transduction and Targeted Therapy 2022 & Bo Meng, nature 2022) reported that Spike of Omicron exhibit weaker binding to ACE2 and Omicron spike protein was less efficiently cleaved and less fusogenicity compared with Delta. According to these evidences, the Omicron variant may infect into host cell through ACE2-independent route. If possible, the authors need to investigate the benefit of sACE2.v2.4-IgG1 in K18-hACE2 transgenic mice inoculated with Omicron BA.1 or BA.2 viruses. At least, discuss this critical issue in discussion section.
2. In Figure 1, the authors used PBS as control, which may not appropriate. A control isotype antibody should be used as a more suitable control.
3. In Figure 1, although sACE2.v2.4-IgG1 inhalation increases survival of gamma infected mice, why almost mice still died at day 10. The authors need to discuss this part.
4. In Figure 1D, the Y-axis may replace to virus titers, which may represent the replication of virus at day 7.
5. How did authors determine the molecule weight of sACE2.v2.4-IgG1? This is important to calculate the dose of four-monomer of ACE2 in molar concentration. The authors perform experiments using molar concentration of ACE2 v2.4-IgG1 and other preclinical antibody, please clarify the Molecular weight of sACE2.v2.4-IgG1 to determine the molar concentration.
6. In methods, dose Spike of BA.2 construct into Myc-pCDNA3.1 vector? In the S binding assay, dose the author measure Spike expressing cell gating the myc positive population by flow cytometry? Please clarify the corrected method.
7. The result of Figure 5 was using structure modeling based on BA.1 RBD with human ACE2. They try to explain why the engineered decoy is broadly active against diverse SARS-CoV-2 variants. But it is crucial to examine the competition ability of sACE2.v2.4-IgG1 is also good enough to disrupt ACE2 receptor and Spike binding.

Referee #2 (Remarks for Author):

The Authors have suggested that sACE2-IgG1 can be used to develop an antiviral against SARS-CoV2. This is an interesting data and the outcome will benefit the development of antiviral against SARS-CoV2. However, the MS suffer from some weakness at this stage.

Major concern:

Figure 1A-B: The level of protection is low (just 20%). The Authors need to investigate if the increased amount of ACE-2 will give better protection ! The authors may also investigate the other administration route!

Figure 1D. What if the Authors analysed the level of vRNA later than 7 days? The administrated ACE-2 may only delayed the replication of virus?

Figure 2D. How author exclude that the effect is not due better binding/neutralization? This need to be demonstrated

Figure 3. What if the authors include Spike proteins with mutation at RBD as control?

Figure 3. It would be interesting to see the data in which authors using higher amount of infectious particles.

Referee #3 (Comments on Novelty/Model System for Author):

The use of P1/gamma strain's in vivo model in this manuscript with considerable focus on omicron strains is somewhat unsatisfactory.

Referee #3 (Remarks for Author):

This manuscript describe results of studies on a previously reported ACE2 decoy, sACE22.v2.4-IgG1, including (1) its administration by inhalation helps protect mice from SARS-CoV2 P1/gamma strain (2) loss of ACE catalytic activity reduces its antiviral effect slightly in the same P1/gamma mouse model (3) sACE22.v2.4-IgG1 binds tightly to S of omicron BA.1 and BA.2 and potently neutralise BA.1.

Although the point (1) is clearly demonstrated with $P=0.0001$ in survival curve, survival difference with and without catalytic activity in point (2) is only barely significant at $P=0.03$. This result was stated as 'supporting dual mechanisms of action: direct blocking of viral S and turnover of ACE2 substrates associated with lung injury and inflammation'. This argument is weak and warrants more supporting evidence in addition to Fig 2B and C showing the same binding activity to the original Wuhan strain, not P1/gamma, so the same experiment needs to be done with P1/gamma's S. In the in vivo study, the levels of viremia and ACE substrates and catabolites should be measured to show no change in virus control and difference in in vivo ACE activity.

The study on omicron strains is very much separate, possibly a stand alone study. Combining this with P1/gamma's in vivo study may give some misleading impression that sACE22.v2.4-IgG1 inhalation is evidenced to be protective. It is important to emphasize that sACE22.v2.4-IgG1 has not been tested against omicron in in vivo models yet. BA.2's neutralisation is important to be demonstrated and not too difficult to perform. The authors may already have some results since submission of this manuscript.

Minor points

Fig 1E cytokine levels: showing levels of uninfected controls may be helpful.

sACE22.v2.4-IgG1 dimer status: it's unclear if they form dimer because of IgG heavy chain dimerization or due to the collectrin-like dimerization domain. Could they be tetramer if both mechanisms effect?

Molecular modelling: the following is stated 'All models are provided in online Supporting Information' but it is unclear which models and in what format. By requesting the information, it turn out that these are 4 PDB files for combination of BA.1 or BA.2 with wild type or v2 ACE2 with no interpretation, while the statement may imply data with all other strains mentioned in the text. What data will be online accessible needs to be clarified in the manuscript main text. Even better more comprehensive analyses of models including earlier strains would be of interest to readers. Indeed new polar contacts are mentioned but more detailed description how these are new and how the earlier strains are different is required.

Fig 5 model; the legend needs to clarify what the pink color indicates, e.g. the tip of N330Y in the third panel.

RESPONSE TO REVIEWERS

Referee #1:

The authors had previously shown that The high-affinity and catalytically active decoy sACE2_{2.v2.4}-IgG1 was effective in vivo against SARS-CoV-2 variants when administered intravenously. In this study, the authors describe the inhalation of sACE2_{2.v2.4}-IgG1 could increase survival and ameliorate lung injury in K18-hACE2 transgenic mice inoculated with a lethal dose of the virulent P.1/gamma virus of decoy ACE2 receptor. The authors also confirm the catalytic activity of ACE2s confers the additional therapeutic benefit. Finally, they proved that ACE2s could tightly bind to omicron BA.1 and BA.2 in vitro binding assay. While the manuscript seems to be ready for publish, but there are some important issues need to be address.

1. The recent articles (Leyun Wu, Signal Transduction and Targeted Therapy 2022 & Bo Meng, nature 2022) reported that Spike of Omicron exhibit weaker binding to ACE2 and Omicron spike protein was less efficiently cleaved and less fusogenicity compared with Delta. According to these evidences, the Omicron variant may infect into host cell through ACE2-independent rout. If possible, the authors need to investigate the benefit of sACE2_{2.v2.4}-IgG1 in in K18-hACE2 transgenic mice inoculated with Omicron BA.1 or BA.2 viruses. At least, discuss this critical issue in discussion section.

We have now tested sACE2_{2.v2.4}-IgG1 in K18-hACE2 transgenic mice inoculated with BA.1 omicron. Consistent with the findings of others, we found that omicron virus infection caused mild disease in this animal model when inoculated at 1×10^4 PFU (see Figure EV1), which is the infectious dose for WA-1/2020 and P.1 variants that causes severe lung injury as we published recently (PMCID: PMC8885411) . We consequently used a very high virus dose (1×10^6 PFU) to induce weight loss and mortality and focus our study on measuring virus replication rather than other metrics of disease/inflammation. We found that sACE2_{2.v2.4}-IgG1 treatment increased mouse survival and protected against weight loss. Viral replication in the lungs was also significantly reduced. The new materials added to the manuscript are as follows (Lines 319-330):

K18-hACE2 transgenic mice were inoculated with SARS-CoV-2 BA.1 omicron virus and treated with isotype-matched IgG1 control or sACE2_{2.v2.4}-IgG1 (15 mg/kg/day IV, once a day for 7 days, beginning 24 h post-inoculation; **Figure 5A**). K18-hACE2 transgenic mice inoculated with SARS-CoV-2 BA.1 omicron at 1×10^4 PFU did not show any weight change (**Figure EV1**), which is consistent with the recent finding that SARS-CoV-2 BA.1 omicron causes mild disease in K18-hACE2 mice (Halfmann *et al*, 2022). Thus, we increased doses 100-fold (1×10^6 PFU) to induce weight loss. 25% of mice in the control group died at Day 10 and the maximal weight loss was up to 32% (**Figure 5B-5D**). Mice treated with sACE2_{2.v2.4}-IgG1 survived to the experiment's conclusion on Day 14 and the maximal weight loss was up to 5% (**Figure 5B-5D**). Levels of viral transcripts for Spike, Nsp, and Rdrp in lung tissue at Day 8 were significantly decreased in treated versus control mice, whereas viral transcript levels were low in both groups at Day 14 when the infection is resolved based on recovery of body weight (**Figure 5E**).

Figure EV1. Body weights of K18-hACE2 transgenic mice inoculated with a low dose of SARS-CoV-2 BA.1 omicron. Normalized body weights of K18-hACE2 mice after intranasal inoculation with 1×10^4 PFU SARS-CoV-2 BA.1 omicron (isolate USA/MD-HP20874/2021). N = 6 mice.

Figure 5. sACE2₂.v2.4-IgG1 inhibits SARS-CoV-2 BA.1 omicron replication in vivo. (A) K18-hACE2 transgenic mice were inoculated intranasally with 1×10^6 PFU SARS-CoV-2 BA.1 omicron (isolate USA/MD-HP20874/2021) and treated 24 h post-inoculation with IgG1 isotype control or sACE2₂.v2.4-IgG1 (15 mg/kg/day intravenously, once per day for 7 days). (B-C) Survival (B) and

weight loss (C). N = 8 (IgG1 control) or 9 (sACE2_{2.v2.4}-IgG1 treated) mice. (D) Maximal weight loss of mice (compared to Day 0) within 14 days following BA.1 omicron virus infection. (E) Viral Spike, Nsp, and RdRp mRNA levels in lung tissue were measured by RT-qPCR at Days 8 and 14, normalized to Ppia transcript levels. Data are presented as mean ± SEM, N = 4 mice per group. ns, not significant; **, p < 0.01 by two-ways ANOVA, corrected by Tukey.

2. In Figure 1, the authors used PBS as control, which may not appropriate. A control isotype antibody should be used as a more suitable control.

It is unclear what the best control is for in vivo studies because the Fc-fused soluble decoy receptor is not strictly an antibody and has shorter half-life (PMCID: PMC8885411). For this reason, we used PBS in all our previous studies. Nonetheless, at the recommendation of the reviewer, we used a nonspecific IgG1 isotype-matched antibody control in the two animal experiments added to the revised manuscript (see Figures 3 and 5). K18-hACE2 mice inoculated with 1×10^4 PFU of SARS-CoV-2 isolate WA-1/2020 and treated with IgG1 control die in 6-7 days (Figure 3D of the revision). In a previous study by our group, mice inoculated with the same infectious dose of WA-1/2020 virus and treated with PBS died in 7-8 days (PMCID: PMC8885411). We consider these differences to be within expected error for independent experiments and conclude that PBS and non-specific isotype-matched antibody are equivalent in this infection model.

3. In Figure 1, although sACE2_{2.v2.4}-IgG1 inhalation increases survival of gamma infected mice, why almost mice still died at day 10. The authors need to discuss this part.

How many mice survive is very much dependent on the specifics: virus dose, virus variant, dose and route of delivery of the test article, by how long treatment is delayed, regimen, etc. For example, we find prophylaxis with sACE2_{2.v2.4}-IgG1 protects all mice from death (PMCID: PMC8885411), yet we see anywhere from all treated mice surviving to 80% death in the three independent animal studies described here (Figures 1B, 2D, 3D, and 5B). We are therefore cautious about discussing what any given survival rate means when so many factors are important. We also note that the dosage and number of doses of inhaled sACE2_{2.v2.4}-IgG1 was considerably lower than what was delivered by IV route. Accordingly, we have added the following sentence to the Discussion on Lines 427-428:

It is possible even greater efficacy may be achieved by increasing the dose amount and frequency.

4. In Figure 1D, the Y-axis may replace to virus titers, which may represent the replication of virus at day 7.

To properly measure virus titer we would need to perform an assay that measures infectious particles, e.g. a plaque assay. While that would certainly be valuable data, we didn't do that here and neither did we have biosafety approval for collecting infectious virus from animal tissues. Instead, we have measured viral RNA transcript levels as a surrogate of virus titer, as is commonly done in the field. We have added the following comment in the Results on Lines 219-223:

Replication of SARS-CoV-2 gamma variant in the lung at Day 7 post-inoculation was measured by reverse transcription and real-time quantitative PCR (RT-qPCR) for the expression levels of

SARS-CoV-2 Spike, non-structural protein (Nsp), and RNA-dependent RNA polymerase (Rdrp) (Figure 1D). Viral transcript levels are expected to correlate with virus titer.

5. How did authors determine the molecule weight of sACE2_{v2.4}-IgG1? This is important to calculate the dose of four- monomer of ACE2 in molar concentration. The authors perform experiments using molar concentration of ACE2 v2.4-IgG1 and other preclinical antibody, please clarify the Molecular weight of sACE2_{v2.4}-IgG1 to determine the molar concentration.

We have added the following clarification to the Methods (Lines 487-490) to avoid confusion:

For consistency, all concentrations in this manuscript are based on monomeric sACE2 subunits (sACE2_{v2.4}-IgG1 has a monomer molecular weight of 108.4 kD) or a H+L chain for antibodies (i.e. concentrations can be considered a measure of binding sites).

6. In methods, dose Spike of BA.2 construct into Myc-pCDNA3.1 vector? In the S binding assay, dose the author measure Spike expressing cell gating the myc positive population by flow cytometry? Please clarify the corrected method.

The reviewer is correct; we first gate the Spike-positive cells based on detection of the myc tag, and then measure the mean fluorescence for the bound ACE2 protein or monoclonal antibody. We have added additional details to the Methods (Lines 521-541) to avoid confusion:

Expi293F cells were transfected with 500 ng plasmid DNA per ml of culture using Expifectamine (Thermo Fisher) according to the manufacturer's directions. After 24-28 h, cells were washed with cold PBS containing 0.2% bovine serum albumin (PBS-BSA) and added to 3-fold serial titrations of the binding proteins in 96-well round-bottomed plates. Plates were incubated on ice for 30 minutes with regular agitation. Cells were washed with PBS-BSA. For assays examining the binding of monomeric sACE2(18-615)-8his, cells were resuspended in PBS-BSA containing 1:150 polyclonal chicken anti-HIS-FITC (Immunology Consultants Laboratory) and 1:300 anti-myc-Alexa 647 (clone 9B11, Cell Signaling Technology). For assays examining the avid binding of dimeric sACE2₂-IgG1 and monoclonal antibodies, cells were resuspended in PBS-BSA containing 1:150 polyclonal chicken anti-MYC-FITC (Immunology Consultants Laboratory) and 1:300 anti-human IgG-APC (clone HP6017, BioLegend). Plates were incubated for 30 minutes on ice with occasional mixing, washed twice with PBS-BSA, and analyzed on a BD Accuri C6 flow cytometer using CFlow version 1.0.264.15. Cells were gated by forward and side scatter to exclude dead cells and debris. Cells expressing Spike were then gated based on detection of the N-terminal myc tag and the binding of sACE2 or monoclonal antibodies was measured for this myc-positive population. No binding of sACE2 or monoclonal antibodies is observed in the Spike/myc-negative population. Binding data are presented as the mean fluorescence units (FITC for bound 8his proteins and APC for bound IgG1 proteins) with subtraction of background fluorescence from cells incubated without sACE2 proteins. Data were normalized across independent experiments based on the total measured fluorescence in each experiment.

7. The result of Figure 5 was using structure modeling based on BA.1 RBD with human ACE2. They try to explain why the engineered decoy is broadly active against diverse SARS-CoV-2

variants. But it is crucial to examine the competition ability of sACE2.v2.4-IgG1 is also good enough to disrupt ACE2 receptor and Spike binding.

For ACE2-based decoys where the binding site on Spike is identical to that for native ACE2 receptors and their binding is mutually exclusive, competition is a function of the decoy's relative affinity/avidity and its concentration. Here, we have high confidence that the decoy effectively competes with native ACE2 due to its tighter affinity and efficacy in neutralizing infection of ACE2-positive cells. We did explicitly test this in two prior publications (PMCID: PMC7888922 and PMCID: PMC7574912), where we used flow cytometry to show that sACE2.v2.4 and wild type sACE2 compete with each other for Spike. We are therefore confident that understanding the molecular basis for the tight affinity of sACE2.v2.4 explains its ability to compete with native receptors and thus neutralize virus infection.

Referee #2:

The Authors have suggested that sACE2-IgG1 can be used to develop an antiviral against SARS-CoV2. This is an interesting data and the outcome will benefit the development of antiviral against SARS-CoV2. However, the MS suffer from some weakness at this stage.

Major concern:

Figure 1A-B: The level of protection is low (just 20%). The Authors need to investigate if the increased amount of ACE-2 will give better protection ! The authors may also investigate the other administration route!

The low level of protection afforded by inhalation is almost certainly due to the low dose. While the amount of protein aerosolized is substantial (~200 mg for each group), the mice only inhale a small fraction of the material. The aerosolized solution enters a holding chamber for the mice and is delivered to the lungs as the mice breath normally. Based on comparisons of prior experiments where we aerosolized protein solution versus direct intratracheal administration (PMCID: PMC8885411), we estimate that in this experiment the mice receive the equivalent of ~0.5 mg/kg dose. Higher doses were not possible due to the amount of protein available. However, we have added an additional in vivo efficacy study where we administered 15 mg/kg/day for 7 days by IV route and observed higher protection (50%). For translational purposes, dosing would of course need to be optimized for human patients but it is also important to keep in mind that a key benefit of the inhalation route we studied in the present manuscript is that it is ideally suited for more frequent dosing in outpatients. Inhalation dosing for other pulmonary diseases such as asthma is readily individualized and we anticipate that inhalation dosing of the decoy will similarly be adjusted for individual patients. The new data is shown in Figure 3 of the revision:

Figure 3. sACE₂.v2.4-IgG1 catalytic activity increases survival of SARS-CoV-2 WA-1/2020 infected K18-hACE2 mice. (A-B) Catalytically active sACE₂.v2.4-IgG1 (A) and catalytically dead sACE₂.v2.4(NN)-IgG1 (B) were immobilized on BLI biosensors and transferred to solutions of RBD (original variant) as the soluble analyte (0-60 s) and returned to buffer to measure dissociation (60-360 s). RBD concentrations are indicated on the right of the BLI sensorgrams. (C) K18-hACE2 transgenic mice were inoculated with 1×10^4 PFU SARS-CoV-2 WA-1/2020 (original variant). Treatment was delayed by 24 hours to allow initial virus replication and evaluate the effects of ACE2 catalytic activity at later stages of disease. sACE₂.v2.4-IgG1 was intravenously administered at 15 mg/kg, once per day for 7 days. (D-E) Survival (D) and weight loss (E) following treatment with an IgG1 isotype control (red) versus catalytically active (blue) and dead (green) sACE₂.v2.4-IgG1. N = 8 mice for IgG1 group and catalytic active group, N = 9 for catalytic dead group. Gehan-Breslow-Wilcoxon test was applied to analyze the survival curves. P = 0.0099 for catalytic active group compared to catalytic dead group. P = 0.017 for catalytic dead group compared to IgG1 control.

Figure 1D. What if the Authors analysed the level of vRNA later than 7 days? The administrated ACE-2 may only delayed the replication of virus?

In a new experiment added to the revision, we infected mice with BA.1 omicron virus and analyzed viral RNA in lung tissue. Groups of N=4 mice were sacrificed at two time points: Day 8 and Day

14. Rather than replication of SARS-CoV-2 being delayed, we found that virus replication had largely ceased by Day 14. The added text (Lines 319-330) and figure are as follows:

K18-hACE2 transgenic mice were inoculated with SARS-CoV-2 BA.1 omicron virus and treated with isotype-matched IgG1 control or sACE2_{2.v2.4}-IgG1 (15 mg/kg/day IV, once a day for 7 days, beginning 24 h post-inoculation; **Figure 5A**). K18-hACE2 transgenic mice inoculated with SARS-CoV-2 BA.1 omicron at 1×10^4 PFU did not show any weight change (**Figure EV1**), which is consistent with the recent finding that SARS-CoV-2 BA.1 omicron causes mild disease in K18-hACE2 mice (Halfmann *et al*, 2022). Thus, we increased doses 100-fold (1×10^6 PFU) to induce weight loss. 25% of mice in the control group died at Day 10 and the maximal weight loss was up to 32% (**Figure 5B-5D**). Mice treated with sACE2_{2.v2.4}-IgG1 survived to the experiment's conclusion on Day 14 and the maximal weight loss was up to 5% (**Figure 5B-5D**). Levels of viral transcripts for Spike, Nsp, and Rdrp in lung tissue at Day 8 were significantly decreased in treated versus control mice, whereas viral transcript levels were low in both groups at Day 14 when the infection is resolved based on recovery of body weight (**Figure 5E**).

Figure 5. sACE2_{2.v2.4}-IgG1 inhibits SARS-CoV-2 BA.1 omicron replication in vivo. (A) K18-hACE2 transgenic mice were inoculated intranasally with 1×10^6 PFU SARS-CoV-2 BA.1 omicron (isolate USA/MD-HP20874/2021) and treated 24 h post-inoculation with IgG1 isotype control or sACE2_{2.v2.4}-IgG1 (15 mg/kg/day intravenously, once per day for 7 days). (B-C) Survival (B) and weight loss (C). N = 8 (IgG1 control) or 9 (sACE2_{2.v2.4}-IgG1 treated) mice. (D) Maximal weight loss of mice (compared to Day 0) within 14 days following BA.1 omicron virus infection. (E) Viral Spike, Nsp, and Rdrp mRNA levels in lung tissue were measured by RT-qPCR at Days 8

and 14, normalized to Ppia transcript levels. Data are presented as mean \pm SEM, N = 4 mice per group. ns, not significant; **, $p < 0.01$ by two-ways ANOVA, corrected by Tukey.

Figure 2D. How author exclude that the effect is not due better binding/neutralization? This need to be demonstrated.

We show that the catalytically dead protein, despite having reduced efficacy in vivo, binds the viral Spike as well as catalytically active protein (Figures 2B-2C and 3A-3B; this includes newer data added to the revision to test affinity to RBD from two variants: P.1/gamma and WA-1/2020). This is consistent with the work of other groups that have made mutations to knock out catalytic activity and find that in vitro inhibition of viral Spike is not diminished (for example, see PMID: PMC8217473 where a disulfide was introduced, and PMC7668070 and PMC8282039 where various point mutations were used, including mutations H374N/H378N used in our study). For the efficacy study in which the decoy proteins were administered by inhalation, we have added viral mRNA measurements from lung tissue showing no significant difference between mice treated with catalytically active versus catalytically dead protein (Figure 2F). Finally, we have added another in vivo study where we administered proteins intravenously after a longer delay following inoculation (new Figure 3; main text description on Lines 262-273). The idea was to better test the proteins' efficacy in ameliorating disease symptoms as opposed to just blocking virus infection early. We observed an even greater effect of catalytic activity in this experiment. Notably, the proteins were purified independently by different scientists for the two experiments and yet we consistently observe that catalytically active protein has greater therapeutic efficacy.

Figure 3. What if the authors include Spike proteins with mutation at RBD as control?

While we didn't test such Spike mutants here, we have explored Spike mutants previously (PMCID: PMC7888922). The ideal Spike mutant control would be one which still interacts with wild type ACE2 receptors to infect cells but is no longer bound by the engineered decoy receptor. A SARS-CoV-2 variant expressing such a Spike would then be resistant to neutralization by the decoy receptor and would beautifully reveal any therapeutic efficacy attributable to ACE2 catalytic activity alone. (Such a Spike mutant would also reveal how SARS-CoV-2 might become partially resistant to the decoy.) However, we failed to find any such mutation in Spike.

Figure 3. It would be interesting to see the data in which authors using higher amount of infectious particles.

Yes it would, and we are also interested in efficacy when treatment is delayed to better mimic treatment of moderate-to-severe COVID for which there are few good drug options. However, we hope the reviewer is understanding that there are only so many in vivo experiments under BL3 containment we can manage. The revised manuscript now has three in vivo experiments supporting our conclusions, summarized in Figures 1, 2D-2F, 3C-3E, and 5, and Extended View Figure EV1. We did use a higher dose of omicron BA.1 virus to induce mortality and weight loss (Figure 5).

Referee #3

The use of P1/gamma strain's in vivo model in this manuscript with considerable focus on omicron strains is somewhat unsatisfactory.

We have added an in vivo assessment of the decoy receptor against an omicron variant. However, the assessment of catalytic activity remains on other variants (Wuhan/original and gamma). Under ideal circumstances we would exhaustively test different proteins against different virus variants using different regimens, but based on the reality of having a certain number of mice available, a certain amount of protein available, and limited expert labor, we faced hard choices to complete the experiments within the time requested by the Editor. We therefore chose to do two new in vivo studies, one in which mice were infected with BA.1 omicron and treated intravenously with sACE2_{2.v2.4}-IgG1, and a second in which mice were infected with WA-1/2020 (an original isolate) and treated intravenously with catalytically active and dead sACE2_{2.v2.4}-IgG1. The reasons for these decisions were as follows:

(i) Transgenic mice infected with omicron variants have attenuated disease. Many groups testing omicron variants in mice do not even report any lung injury or mortality, and instead use omicron infection of mice as a model for virus replication in vivo but not of disease. We increased the virus dose by 2 orders of magnitude to induce weight loss in infected K18-hACE2 mice (see Figure EV1 and Figure 5), but even so only 25% of infected animals died and death was delayed until Day 10. We therefore did not consider omicron infected mice suitable for evaluating the effect of ACE2 catalytic activity on ameliorating disease, especially when the effect of knocking out catalytic activity was small in our original experiment.

(ii) To better test any therapeutic contribution of ACE2 catalytic activity, we infected K18-hACE2 mice with WA-1/2020 virus, which induces severe lung vascular leakage, lung edema, and death (PMCID: PMC8885411). Treatment was begun after 24 hours, thus allowing the virus to begin replication. Anecdotally, at 24 hours the mice appear sick. We thus consider this infection protocol and treatment regimen a robust test of whether sACE2_{2.v2.4}-IgG1 might contribute to reducing disease symptoms as opposed to simply blocking virus neutralization. The results showed that 89% of mice treated with catalytically dead sACE2_{2.v2.4}(NN)-IgG1 died with extended survival by ~1 day compared to the control group, whereas 50% of mice treated with catalytically active sACE2_{2.v2.4}-IgG1 survived. While the WA-1/2020 variant is no longer representative of the SARS-CoV-2 variants currently in circulation, we believe that evaluating therapeutic efficacy in an animal model that mimics COVID symptoms is of greater importance if the objective is to develop drugs for hospitalized patients where there is still a clinical need.

(iii) We administered protein by IV instead of inhalation. Aerosolizing protein solution into a holding chamber (from which the mice inhale just a small fraction) requires a very large amount of material and limits what doses can be delivered.

While more in vivo studies would better address all the reviewer's comments, to accomplish the additions for this revision we exhausted all the K18-hACE2 mice we had available. We do hope the reviewer will be pleased with what we are able to provide, all of which supports our original conclusions.

This manuscript describe results of studies on a previously reported ACE2 decoy, sACE2_{2.v2.4}-IgG1, including (1) its administration by inhalation helps protect mice from SARS-CoV2 P1/gamma strain (2) loss of ACE catalytic activity reduces its antiviral effect slightly in the same P1/gamma mouse model (3) sACE2_{2.v2.4}-IgG1 binds tightly to S of omicron BA.1 and BA.2 and potently neutralise BA.1.

Although the point (1) is clearly demonstrated with P=0.0001 in survival curve, survival difference with and without catalytic activity in point (2) is only barely significant at P=0.03. This result was stated as 'supporting dual mechanisms of action: direct blocking of viral S and turnover of ACE2 substrates associated with lung injury and inflammation'. This argument is weak and warrants more supporting evidence in addition to Fig 2B and C showing the same binding activity to the original Wuhan strain, not P1/gamma, so the same experiment needs to be done with P1/gamma's S. In the in vivo study, the levels of viremia and ACE substrates and catabolites should be measured to show no change in virus control and difference in in vivo ACE activity.

We have added another in vivo experiment to compare catalytically active sACE2_{2.v2.4}-IgG1 to catalytically dead sACE2_{2.v2.4(NN)}-IgG1. To better observe the possible effects of eliminating ACE2 catalytic activity on treating disease, K18-hACE2 mice were infected with a lethal dose of an original virus variant (WA-1/2020) that is associated with severe lung injury. Treatment was begun 24 hours later so as to not immediately neutralize virus infection. Using this regimen, we found that the catalytic dead protein was markedly less effective at protecting the mice from death. The new additions to the manuscript (Lines 262-273) are:

To further evaluate the therapeutic contributions of ACE2 catalytic activity, K18-hACE2 transgenic mice were intranasally inoculated with a lethal dose (1×10^4 PFU) of SARS-CoV-2 isolate USA-WA1/2020 (an original variant) and protein administration was delayed for 24 h to allow initial virus replication (**Figure 3C**). This mouse model of COVID-19 induces severe lung injury and edema (Zhang *et al*, 2022) and was thus considered a challenging test for how catalytic activity may ameliorate disease pathology. Inoculated mice (8/9 per group) were treated with 15 mg/kg/day for 7 days intravenously (**Figure 3C**). All mice administered an unrelated isotype-matched IgG1 as a control died within 6-7 days, whereas 50% of mice treated with catalytically active sACE2_{2.v2.4}-IgG1 survived (**Figure 3D and 3E**). These results closely match prior published data in which PBS was administered as the control (Zhang *et al*, 2022). By comparison, catalytically dead sACE2_{2.v2.4(NN)}-IgG1 prolonged survival by just ~1 day with a low survival rate of 11% (**Figure 3D and 3E**).

Figure 3. sACE₂.v2.4-IgG1 catalytic activity increases survival of SARS-CoV-2 WA-1/2020 infected K18-hACE2 mice. (A-B) Catalytically active sACE₂.v2.4-IgG1 (A) and catalytically dead sACE₂.v2.4(NN)-IgG1 (B) were immobilized on BLI biosensors and transferred to solutions of RBD (original variant) as the soluble analyte (0-60 s) and returned to buffer to measure dissociation (60-360 s). RBD concentrations are indicated on the right of the BLI sensorgrams. (C) K18-hACE2 transgenic mice were inoculated with 1×10^4 PFU SARS-CoV-2 WA-1/2020 (original variant). Treatment was delayed by 24 hours to allow initial virus replication and evaluate the effects of ACE2 catalytic activity at later stages of disease. sACE₂.v2.4-IgG1 was intravenously administered at 15 mg/kg, once per day for 7 days. (D-E) Survival (D) and weight loss (E) following treatment with an IgG1 isotype control (red) versus catalytically active (blue) and dead (green) sACE₂.v2.4-IgG1. N = 8 mice for IgG1 group and catalytic active group, N = 9 for catalytic dead group. Gehan-Breslow-Wilcoxon test was applied to analyze the survival curves. P = 0.0099 for catalytic active group compared to catalytic dead group. P = 0.017 for catalytic dead group compared to IgG1 control.

We have also added affinity measurements for the catalytically active and dead proteins to the RBD of P.1/gamma virus. The data are provided in Figures 2B and 2C.

For the inhalation study, we did have groups of mice (N = 4 per group) that were sacrificed on Day 7 to measure transcript levels in lung tissue. We have added the data for viral transcript

levels for animals treated with catalytically active and dead proteins to Figure 2F. No significant difference in viral load was observed between these groups. We have added the following sentence on Lines 258-260:

However, no difference was observed in viral load in the lungs at Day 7 between the two groups (**Figure 2F**), consistent with catalytically dead sACE₂.v2.4(NN)-IgG1 neutralizing virus despite having reduced therapeutic efficacy.

Figure 2. Catalytic activity of sACE₂.v2.4-IgG1 contributes to the therapeutic efficacy to mitigate mouse lung injury and improve survival following SARS-CoV-2 gamma infection. (A) A 7-Methoxycoumarin-4-acetyl (MCA) conjugated peptide is quenched by a 2,4-dinitrophenyl group. ACE2 catalyzed cleavage of the peptide is measured by increased MCA fluorescence. Mutations H374N and H378N generate a catalytically dead sACE₂.v2.4-IgG1 protein. (B-C) Catalytically active sACE₂.v2.4-IgG1 (B) and catalytically dead sACE₂.v2.4(NN)-IgG1 (C) were immobilized on BLI biosensors that were transferred to solutions of gamma RBD as the soluble analyte (0-60 s) and returned to buffer to measure dissociation (60-360 s). RBD

concentrations are indicated on the right of the sensorgrams. **(D-E)** Catalytically active sACE2_{2.v2.4}-IgG1 and catalytically dead sACE2_{2.v2.4(NN)}-IgG1 were aerosolized (7.5 ml protein at 8.3 mg/ml in 25 minutes) and delivered by inhalation to K18-hACE2 transgenic mice at 12 h, 48 h, and 84 h post-inoculation with SARS-CoV-2 gamma variant. 10 mice in each group were observed for survival **(D)** and weight loss **(E)**. The P-value of survival curve by Gehan-Breslow-Wilcoxon test is shown. Catalytically active and inactive proteins were tested in the same experiment versus PBS control shown in Figure 1. **(F)** Viral load in the lung was measured by RT-qPCR at Day 7. The mRNA expression levels of SARS-CoV-2 Spike, Nsp, and Rdrp are normalized to Ppia. Data are presented as mean ± SEM, N = 4 mice per group. ns, no significance by unpaired Student's t-test with two sided.

We are unable to provide measurements of angiotensin peptides. To properly assess the low levels (picograms per ml of serum or plasma) of potentially active yet labile angiotensin peptides requires careful sample collection to immediately inhibit protease activity and ideally LC-MS methods. (While there are commercial ELISAs for angiotensin peptides, their accuracy is questioned as they have given conflicting data [PMCID: PMC7878344].) While we are very much interested in understanding how angiotensin peptides change during the course of SARS-CoV-2 infection, with and without treatment, this will require method development, method approval for handling samples from infected animals, and detailed assessments over time in many animals. It also isn't clear to us that there should be no difference in angiotensin peptides in control infected animals versus animals treated with catalytic dead sACE2, since neutralizing virus replication in-and-of-itself is expected to impact endogenous ACE2 levels in the infected animals.

We have previously shown that administration of catalytically active sACE2_{2.v2.4}-IgG1 is associated with a large increase in ACE2 catalytic activity in mouse plasma. We have added the following to the text on Lines 238-240:

We have previously shown that administration of sACE2_{2.v2.4}-IgG1 to mice is associated with large increases in ACE2 catalytic activity detected in plasma (Zhang *et al*, 2022). To test whether the decoy's proteolytic activity contributes to the mechanism of reducing disease following SARS-CoV-2 infection...

The study on omicron strains is very much separate, possibly a stand alone study. Combining this with P1/gamma's in vivo study may give some misleading impression that sACE2_{2.v2.4}-IgG1 inhalation is evidenced to be protective. It is important to emphasize that sACE2_{2.v2.4}-IgG1 has not been tested against omicron in in vivo models yet. BA.2's neutralisation is important to be demonstrated and not too difficult to perform. The authors may already have some results since submission of this manuscript.

We have now added neutralization of BA.2 omicron pseudovirus and an in vivo efficacy study in mice infected with BA.1 omicron. To ensure our work is as relevant as possible to a constantly changing pandemic, we have also added binding data to BA.4/BA.5 omicron Spike. The additions to the manuscript are as follows:

Lines 319-330: K18-hACE2 transgenic mice were inoculated with SARS-CoV-2 BA.1 omicron virus and treated with isotype-matched IgG1 control or sACE2_{2.v2.4}-IgG1 (15 mg/kg/day IV, once

a day for 7 days, beginning 24 h post-inoculation; **Figure 5A**). K18-hACE2 transgenic mice inoculated with SARS-CoV-2 BA.1 omicron at 1×10^4 PFU did not show any weight change (**Figure EV1**), which is consistent with the recent finding that SARS-CoV-2 BA.1 omicron causes mild disease in K18-hACE2 mice (Halfmann *et al*, 2022). Thus, we increased doses 100-fold (1×10^6 PFU) to induce weight loss. 25% of mice in the control group died at Day 10 and the maximal weight loss was up to 32% (**Figure 5B-5D**). Mice treated with sACE2₂.v2.4-IgG1 survived to the experiment's conclusion on Day 14 and the maximal weight loss was up to 5% (**Figure 5B-5D**). Levels of viral transcripts for Spike, Nsp, and Rdrp in lung tissue at Day 8 were significantly decreased in treated versus control mice, whereas viral transcript levels were low in both groups at Day 14 when the infection is resolved based on recovery of body weight (**Figure 5E**).

Lines 334-344: The S sequences of BA.1 and BA.2 are separated by 26 mutations and may therefore differ substantially in their interactions with binding proteins. BA.4 and BA.5 omicron variants have the same S sequences and differ from BA.2 at 5 positions. Using flow cytometry to measure binding of monomeric sACE2(18-615) proteins to BA.2 or BA.4/BA.5 S-expressing cells, it was observed that engineered decoys carrying the v2.4 or CDY14 mutations bound tighter than wild type sACE2 (**Figure 6A and 6D**). Furthermore, avid binding of sACE2₂.v2.4-IgG1 to BA.2 or BA.4/BA.5 omicron S-expressing cells outperformed four monoclonal antibodies authorized for emergency use as therapeutics in nearly all cases (**Figure 6B and 6E**). In HeLa cells expressing human ACE2 that were infected with a BA.2 omicron pseudovirus carrying a luciferase reporter gene, sACE2₂.v2.4-IgG1 inhibited virus replication with an IC₅₀ of 130 ± 40 pM compared to 350 ± 90 pM for wild type sACE2₂-IgG1 (**Figure 6C**).

Figure 5. sACE2₂.v2.4-IgG1 inhibits SARS-CoV-2 BA.1 omicron replication in vivo. (A) K18-hACE2 transgenic mice were inoculated intranasally with 1×10^6 PFU SARS-CoV-2 BA.1 omicron (isolate USA/MD-HP20874/2021) and treated 24 h post-inoculation with IgG1 isotype control or sACE2₂.v2.4-IgG1 (15 mg/kg/day intravenously, once per day for 7 days). (B-C) Survival (B) and weight loss (C). N = 8 (IgG1 control) or 9 (sACE2₂.v2.4-IgG1 treated) mice. (D) Maximal weight loss of mice (compared to Day 0) within 14 days following BA.1 omicron virus infection. (E) Viral Spike, Nsp, and Rdrp mRNA levels in lung tissue were measured by RT-qPCR at Days 8 and 14, normalized to Ppia transcript levels. Data are presented as mean \pm SEM, N = 4 mice per group. ns, not significant; **, $p < 0.01$ by two-ways ANOVA, corrected by Tukey.

Figure 6. The engineered decoy tightly binds S of BA.2 and BA.4/BA.5 omicron. (A) Using flow cytometry, binding to Expi293F cells expressing S of BA.2 omicron was measured for monomeric sACE2-8his proteins. Data are mean \pm SEM, N = 3 independent experiments. (B) Binding of antibodies and dimeric sACE2₂-IgG1 proteins to Expi293F cells expressing BA.2 omicron S measured by flow cytometry. Data are mean \pm SEM, N = 4 independent experiments. (C) BA.2 omicron pseudovirus was incubated with sACE2₂-IgG1 (grey) or sACE2₂.v2.4-IgG1 (blue) for 1 h and added to HeLa-hACE2-11 cells. Infection was measured after 48 h. Data are mean \pm SD, N = 3 independent replicates. (D-E) Binding of monomeric sACE2-8his proteins (D) and dimeric IgG1 proteins (E) to Expi293F cells expressing BA.4/BA.5 omicron S as measured by flow cytometry. Data are mean \pm SEM, N = 3 independent experiments.

Minor points

Fig 1E cytokine levels: showing levels of uninfected controls may be helpful.

Cytokine transcript levels were measured in uninfected control animals and these data are now provided in Extended View Table EV1. We have added the following sentence to the main text on lines 227-228:

Aerosol delivery of sACE2.v2.4-IgG1 reduced cytokine expression in the lungs, although the levels remained higher than in uninfected mice (Table EV1).

Table EV1. Cytokine expression (transcript levels relative to Ppia).			
Gene	Uninfected and untreated	Day 7 post-inoculation	
		PBS	sACE2.v2.4-IgG1
Il1a	0.0379	0.1985	0.1648
	0.0105	0.1562	0.0475
	0.0051	0.4054	0.067
	0.0079	0.3578	0.0165
Il1b	0.0076	0.067	0.0248
	0.0094	0.0496	0.0183
	0.0085	0.1369	0.0126
	0.0026	0.1081	0.058
Tnfa	0.0070	0.0854	0.0434
	0.0031	0.1345	0.0701
	0.0055	0.0595	0.0468
	0.0050	0.1083	0.026
Ifng	0.0016	0.0095	0.0059
	0.0003	0.0108	0.0024
	0.0013	0.0089	0.0092
	0.0009	0.0118	0.0052

Lung tissue was harvested from N = 4 mice in each group. Expression levels based on RT-qPCR are listed for each animal.

sACE2.v2.4-IgG1 dimer status: it's unclear if they form dimer because of IgG heavy chain dimerization or due to the collectrin-like dimerization domain. Could they be tetramer if both mechanisms effect?

Excellent question since both the Fc moiety by itself and sACE2 by itself form dimers. We previously addressed this very question by analytical size exclusion chromatography (PMCID: PMC7574912) and found that the protein forms a stable dimer overall. Other groups working on sACE2-Fc fusions also observe only dimeric protein.

Molecular modelling: the following is stated 'All models are provided in online Supporting Information' but it is unclear which models and in what format. By requesting the information, it turn out that these are 4 PDB files for combination of BA.1 or BA.2 with wild type or v2 ACE2 with no interpretation, while the statement may imply data with all other strains mentioned in the text. What data will be online accessible needs to be clarified in the manuscript main text. Even better more comprehensive analyses of models including earlier strains would be of interest to

readers. Indeed new polar contacts are mentioned but more detailed description how these are new and how the earlier strains are different is required.

We have modified this section with new detail, in addition to adding material regarding BA.4/BA.5 omicron against which the decoy maintains high affinity (see Figures 6D and 6E). We previously did extensive MD simulations of wild type ACE2 and engineered ACE2.v2.4 bound to the RBDs of three virus variants (Wuhan, delta, and gamma); that work was on the micro- to milli-second time scale, was achieved using enormous computational resources, and was analyzed in detail in a prior publication (PMCID: PMC7574912). The objective of the current modeling was to specifically see why ACE2.v2.4 binds with tight affinity to omicron RBD, despite all the changes in the virus sequence. The answer is essentially the same as what we previously observed for the other variants by MD; some new atomic contacts are formed and changes in the RBD sequence do not introduce any unresolvable clashes at the ACE2.v2.4 interface. It is fair to say that this section therefore doesn't add much new material beyond what we have previously described for other virus variants. We have revised this section to better communicate that the findings we are reporting based on modeling omicron RBD bound to ACE2 proteins are entirely consistent with simulations of earlier virus variants (Lines 348-391):

We previously used molecular dynamics simulations on the micro- to milli-second time scale to understand why ACE2-based decoys carrying the v2.4 mutations bind much tighter to S from the Wuhan, delta, and gamma variants (Zhang *et al*, 2022). Compared to wild type ACE2, ACE2.v2.4 was found to make new atomic contacts to the RBD and is 'pre-stabilized' in the RBD-bound conformation. Here, to understand why soluble derivatives of ACE2.v2.4 continue to bind with higher affinity than wild type ACE2 to the RBD of omicron variants, we modeled the interacting proteins using ROSETTA (Leman *et al*, 2020). The cryo-electron microscopy structure of BA.1 omicron RBD bound to ACE2 (PDB 7WPB, 2.79 Å resolution) was used as a template for modeling ACE2.v2.4 bound to BA.1, BA.2, and BA.4/BA.5 RBDs (Yin *et al*, 2022). Structures were relaxed using the ROSETTA energy function (see Methods) and PDB files for all models are provided in online Supporting Information.

BA.1 and BA.2 omicron RBDs have identical residues at the interface except at position 496 (serine in BA.1 and glycine in BA.2), which is 7.6 Å from ACE2-D38 (C α -C α distance). Due to their close similarity at the interface, we describe here the BA.1 omicron models as representative of atomic interactions by ACE2.v2.4 to both BA.1 and BA.2 omicron. Substitution T27Y in ACE2.v2.4 brings the aromatic ring of tyrosine-27 into a cluster of hydrophobic residues on omicron formed by RBD-F456, Y473, A475, and Y489 (**Figure 7A**). This is associated with minor backbone movements of RBD loop 1 (a.a. 455-491) and a shift of RBD-Y473 to resolve a small steric clash with the larger ACE2-T27Y side chain. We speculate that dynamic flexibility of RBD loops accommodates mutations on the ACE2 surface to resolve minor clashes. The two other v2.4 mutations, ACE2-L79T and ACE2-N330Y, are at the interface periphery. RBD-F486 makes contacts to ACE2-L79T (**Figure 7B**), while RBD-T497 of RBD loop 2 (a.a. 496-506) moves closer to pack against ACE2-N330Y (**Figure 7C**). ACE2.v2.4 forms new polar contacts between the side chain hydroxyls of ACE2-T27Y and RBD-Y473 and between the hydroxyl of ACE2-N330Y and backbone carbonyl of RBD-P499 that are not seen in the complexes with wild type ACE2 (compare the wild type/grey and ACE2.v2.4/colored structures in **Figure 7A and 7C**). The new atomic contacts formed, together with increased hydrophobic packing around ACE2-

T27Y, may all contribute to the enhanced affinity of ACE2.v2.4 and were observed and described in our prior molecular dynamics-based modeling of ACE2.v2.4 bound to the RBDs of SARS-CoV-2 Wuhan, delta, and gamma variants (Zhang *et al*, 2022), suggesting that the atomic mechanisms of enhanced affinity are shared across all the virus variants.

BA.4/BA.5 omicron S differs from the other omicron sublineages at two sites in the interface with ACE2. First, there is a glutamine at RBD position 493 (BA.1 and BA.2 both have arginine); this reverts position 493 back to the amino acid found in the original SARS-CoV-2 S sequence against which ACE2.v2.4 was engineered for enhanced affinity. Second, RBD-F486 is mutated to valine and is modeled to make direct atomic contacts with ACE2-L79 in wild type ACE2 (RBD-486V C γ 2 is 3.8 Å from ACE2-L79 C δ 2) or with ACE2-L79T in the engineered ACE2.v2.4 decoy (RBD-486V C γ 1 and C γ 2 are 3.9 Å and 3.8 Å from ACE2-79T C γ 2) (**Figure 7D**). Consequently, modeling predicts that the RBD-F486V substitution interacts favorably with both wild type receptors and the engineered decoy, consistent with sACE2_{v2.4}-IgG1 maintaining high affinity to BA.4/BA.5 omicron (**Figure 6E**).

Figure 7. Molecular basis for enhanced affinity of the engineered decoy for omicron RBD. (A-C) The BA.1 omicron RBD (dark red for carbon atoms, CPK coloring for non-carbon atoms) bound to wild type ACE2 (grey) and engineered ACE2.v2.4 (blue for carbon atoms, CPK coloring for non-carbon atoms) was modeled using ROSETTA. Superpositions of the models are shown in the regions surrounding ACE2.v2.4 mutations T27Y (A), L79T (B), and N330Y (C). New polar contacts formed by the ACE2.v2.4 mutations are indicated with dashed black lines. (D) Model of BA.4/BA.5 omicron RBD (orange for carbon atoms, CPK coloring for non-carbon atoms) bound to wild type ACE2 (grey) and engineered ACE2.v2.4 (blue for carbon atoms, CPK coloring for non-carbon atoms). The region surrounding ACE2.v2.4 mutation L79T is shown.

ONLINE SUPPORTING INFORMATION

PDB files are provided of the following ROSETTA models:

BA.1 omicron RBD bound to the protease domain of wild type ACE2

BA.1 omicron RBD bound to the protease domain of ACE2.v2.4

BA.2 omicron RBD bound to the protease domain of wild type ACE2

BA.2 omicron RBD bound to the protease domain of ACE2.v2.4

BA.4/BA.5 omicron RBD bound to the protease domain of wild type ACE2

BA.4/BA.5 omicron RBD bound to the protease domain of ACE2.v2.4

Fig 5 model; the legend needs to clarify what the pink color indicates, e.g. the tip of N330Y in the third panel.

We have clarified that non-carbon atoms in the ACE2.v2.4-bound RBD structure are in CPK colors, where oxygen is red (what the reviewer saw as pink), nitrogen is blue, and sulfur is yellow.

30th Aug 2022

Dear Dr. Procko,

Thank you for the submission of your revised manuscript to EMBO Molecular Medicine. I am pleased to inform you that we will be able to accept your manuscript pending the following final amendments:

- 1) We note that you currently have together with you, a total of 5 co-corresponding authors. Is that correct? While there is no limit per se to the number of co-corresponding authors, 5 is rare, and may not reflect as intended to the community. Do you confirm equal contribution of these 5 people, able to take full responsibility for the paper and its content?
- 2) In the main manuscript file, please do the following:
 - Correct/answer the track changes suggested by our data editors by working from the attached document.
 - Add up to 5 keywords.
 - Figures should be called out in a sequential order. Currently Figure 3A and B are called out before Figure 2D. Please correct.
 - Rename Tables EV1 and EV2 to Tables 1 and 2 and correct their callouts in the main text.
 - In M&M, please specify the biosafety level for the experiments with SARS-CoV-2 by adding and amending the following sentence: All experiments with SARS-CoV-2 were performed in a ... level laboratory and with approval from...
 - In M&M, add statistical paragraph that should reflect all information that you have filled in the Authors Checklist, especially regarding randomization, blinding, replication.
 - Data availability: If no data are deposited in public repositories, please add the sentence: "This study includes no data deposited in external repositories".

Please check "Author Guidelines" for more information.

<https://www.embopress.org/page/journal/17574684/authorguide#availabilityofpublishedmaterial>

- 3) Online supporting information: Please remove this section. Information provided here could either be moved to "Data availability section" or simply provided as source data as you already did. If you decide to move it to "Data availability section" please use following format:

[data type]: [full name of the resource] [accession number/identifier] ([doi or URL or identifiers.org/DATABASE:ACCESSION])

Please check "Author Guidelines" for more information.

<https://www.embopress.org/page/journal/17574684/authorguide#availabilityofpublishedmaterial>

- 4) Source data: Please zipp all files corresponding to Figure 7 and submit one file per figure. Please check "Author Guidelines" for more information. <https://www.embopress.org/page/journal/17574684/authorguide#sourcedata>
- 5) Conflict of interest: Rename "Conflict of interest" to "Disclosure Statement & Competing Interests". We updated our journal's competing interests policy in January 2022 and request authors to consider both actual and perceived competing interests. Please review the policy <https://www.embopress.org/competing-interests> and update your competing interests if necessary.
- 6) Synopsis:
 - Synopsis text: Please remove it from the main text and upload a separate .doc file.
 - Synopsis image: Please provide a striking image or visual abstract as a high-resolution jpeg file 550 px-wide x (250-400)-px high to illustrate your article.
 - Please check your synopsis text and image and submit their final versions with your revised manuscript. Please be aware that in the proof stage minor corrections only are allowed (e.g., typos).
- 7) For more information: This space should be used to list relevant web links for further consultation by our readers. Could you identify some relevant ones and provide such information as well? Some examples are patient associations, relevant databases, OMIM/proteins/genes links, author's websites, etc...
- 8) Press release: Please inform us as soon as possible and latest at the time of submission of the revised manuscript if you plan a press release for your article so that our publisher could coordinate publication accordingly.
- 9) Please be aware that we use a unique publishing workflow for COVID-19 papers: a non-typeset PDF of the accepted manuscript is published as "Just Accepted" on our website. With respect to a possible press release, we have the option to not post the "Just Accepted" version if you prefer to wait with the press release for the typeset version. Please let us know whether you agree to publication of a "Just accepted" version or you prefer to wait for the typeset version.
- 10) As part of the EMBO Publications transparent editorial process initiative (see our Editorial at <http://embomolmed.embopress.org/content/2/9/329>), EMBO Molecular Medicine will publish online a Review Process File (RPF) to accompany accepted manuscripts. This file will be published in conjunction with your paper and will include the anonymous referee reports, your point-by-point response and all pertinent correspondence relating to the manuscript. Let us know whether you agree with the publication of the RPF and as here, if you want to remove or not any figures from it prior to publication. Please note that the Authors checklist will be published at the end of the RPF.
- 11) Please provide a point-by-point letter INCLUDING my comments as well as the reviewer's reports and your detailed responses (as Word file).

I look forward to reading a new revised version of your manuscript as soon as possible.

Yours sincerely,

Zeljko Durdevic

*

***** Reviewer's comments *****

Referee #1 (Remarks for Author):

The authors have adequately responded to the comments.

Referee #2 (Comments on Novelty/Model System for Author):

This work demonstrate that using recombinant ACE-2 (by using Inhalation) have an good antiviral efficacy.

Referee #2 (Remarks for Author):

Th authors have addressed my concerns.

Referee #3 (Remarks for Author):

The manuscript has been improved satisfactorily.

The authors performed the requested editorial changes. All 5 co-corresponding authors made substantial and critical contributions to the research project. All 5 co-corresponding authors take full responsibility for the paper and its content.

We are pleased to inform you that your manuscript is accepted for publication and is now being sent to our publisher to be included in the next available issue of EMBO Molecular Medicine.